# Ferromagnetic quantum critical point protected by nonsymmorphic symmetry in a Kondo metal

Soohyeon Shin [1,4] ✉, Aline Ramires [2,4] ✉, Vladimir Pomjakushin[1,3], Igor Plokhikh[1] & Ekaterina Pomjakushina [1]

Quantum critical points (QCPs), zero-temperature phase transitions, are windows to fundamental quantum-mechanical phenomena associated with universal behaviour. Magnetic QCPs have been extensively investigated in the vicinity of antiferromagnetic order. However, QCPs are rare in metallic ferromagnets due to the coupling of the order parameter to electronic soft modes. Recently, antisymmetric spin-orbit coupling in noncentrosymmetric systems was suggested to protect ferromagnetic QCPs. Nonetheless, multiple centrosymmetric materials host FM QCPs, suggesting a more general mechanism behind their protection. In this context, $CeSi_{2-\delta}$, a dense Kondo lattice crystallising in a centrosymmetric structure, exhibits ferromagnetic order when Si is replaced with Ag. We report that the Ag-substitution to $CeSi_{1.97}$ linearly suppresses the ferromagnetic order towards a QCP, accompanied by concurrent strange-metal behaviour. Herein, we suggest that, despite the centrosymmetric structure, spin-orbit coupling arising from the local noncentrosymmetric structure, in combination with nonsymmorphic symmetry, can protect ferromagnetic QCPs. Our findings offer a general guideline for discovering new ferromagnetic QCPs and highlight one new family of materials within which the interplay of topology and quantum phase transitions can be investigated in the context of strongly correlated systems.

Quantum critical points (QCP), zero-Kelvin second-order phase transitions, are known for promoting exotic quantum phenomena associated with strong order parameter fluctuations[1,2]. The observation of unconventional superconductivity and strange-metal behaviour around QCPs has led to intense research over the past decades, but important questions remain to be addressed. One of the pressing questions in the field concerns the origin and nature of the strange metal or non-Fermi liquid (NFL) behaviour associated with QCPs. NFL behaviour has been reported in multiple materials in the family of heavy fermions, spanning very different chemical compositions and crystalline symmetries.

Nevertheless, quantum critical behaviour seems to be universal, with recurrent experimental signatures such as linear temperature dependence of resistivity and logarithmic temperature dependence of specific heat. This universal behaviour seems to dominate over the specific details of the lattice structure, in agreement with the theoretical understanding that critical phenomena depend primarily on long-wavelength and low-frequency excitations. Nevertheless, topological electronic systems, for which lattice symmetries are key, challenge this understanding and have reopened questions associated with the role of lattice symmetries on the nature of quantum criticality[3].

[1]Laboratory for Multiscale Materials Experiments (LMX), PSI Center for Neutron and Muon Sciences, Paul Scherrer Institut, CH-5232 Villigen PSI, Switzerland. [2]Laboratory for Theoretical and Computational Physics (LTC), PSI Center for Scientific Computing, Theory and Data, Paul Scherrer Institut, CH-5232 Villigen PSI, Switzerland. [3]Laboratory for Neutron Scattering and Imaging (LNS), PSI Center for Neutron and Muon Sciences, Paul Scherrer Institut, CH-5232 Villigen PSI, Switzerland. [4]These authors contributed equally: Soohyeon Shin, Aline Ramires. ✉e-mail: soohyeon.shin@psi.ch; aline.ramires@psi.ch

The universality of quantum critical phenomena is also known to be associated with the nature of the order parameter suppressed at the QCP. While universal NFL behaviour has been investigated around various antiferromagnetic (AFM) QCPs[4], ferromagnetic (FM) QCPs are scarce, as they are often cut out by a first-order transition[5] or smeared out by inhomogeneity[6] or short-range order[7]. Recently, QCPs were reported in the itinerant FMs $UCo_{1-x}Fe_xGe$[8] and $Mn_{1-x}Cr_xSi_2$[9]. Although rare, local-moment FM QCPs with concurrent NFL behaviour have been reported in clean systems such as $CeRh_6Ge_4$ and $YbNi_4(P_{1-x}As_x)_2$[2,10,11]. Theoretically, the protection of the FM QCP in the latter has been ascribed to low dimensionality, spin anisotropy, or the notion of local criticality.

The mechanism behind the transmutation of FM QCP into first-order quantum phase transitions (QPTs) is based on the presence of soft electronic modes, massless two-particle excitations, of a generic Fermi liquid when spin-orbit coupling (SOC) is negligible[12,13]. The diverging spin susceptibility of the Fermi liquid at zero temperature and magnetic field contributes to the free energy with non-analytic terms in the magnetisation, making the phase transition first-order at low temperatures[12–14]. However, it has been suggested that strong spin-orbit coupling introduced by the breaking of spatial inversion symmetry in noncentrosymmetric materials introduces a cut-off to this divergence and can restore the second-order nature of the QPT[15]. One material in which this mechanism might be at play is $CeRh_6Ge_4$, which has a noncentrosymmetric structure[11]. However, $YbNi_4(P_{1-x}As_x)_2$ also hosts an FM QCP but adopts a centrosymmetric structure[10]. More examples of materials displaying FM QCPs are, therefore, highly desirable in order to unravel the mechanisms behind their manifestation.

$CeSi_{2-\delta}$ is the first dense Kondo FM in which ordered Ce moments are reduced due to Kondo screening. Figure 1a shows that Ce and Si atoms are located at 4a(0,3/4,1/8) and 8e(0, 1/4, 0.2910(2)) Wyckoff position, respectively, in the centrosymmetric $ThSi_2$-type tetragonal structure ($I4_1/amd$, no. 141)[16]. Stoichiometric $CeSi_2$ exhibits a paramagnetic ground state, with the Ce moments Kondo-screened by the conduction electrons, as schematically depicted on the left of Fig. 1d[17]. When Si-site deficiency ($\delta$) is larger than 0.15, the Ce moments order ferromagnetically below $T = 11$–14 K, with the Curie temperature increasing with $\delta$[18]. In addition, early pressure experiments on $CeSi_{1.81}$ suggested the existence of an FM QCP near $P = 13.1$ kbar[19]. However, the concurrent NFL behaviour associated with the $\delta$-driven putative FM QCP has not been investigated due to difficulties related to a structural transition and thermal expansion[18]. Nonetheless, it has been reported that replacing Si with Ag also induces FM ordering without any structural transition[16]. Here, we report a potential FM QCP and concurrent NFL behaviour by controlling Ag-doping in the dense Kondo lattice $CeSi_{2-\delta}$ and suggest the importance of local non-centrosymmetricity and nonsymmorphicity for the general stabilisation of FM QCPs.

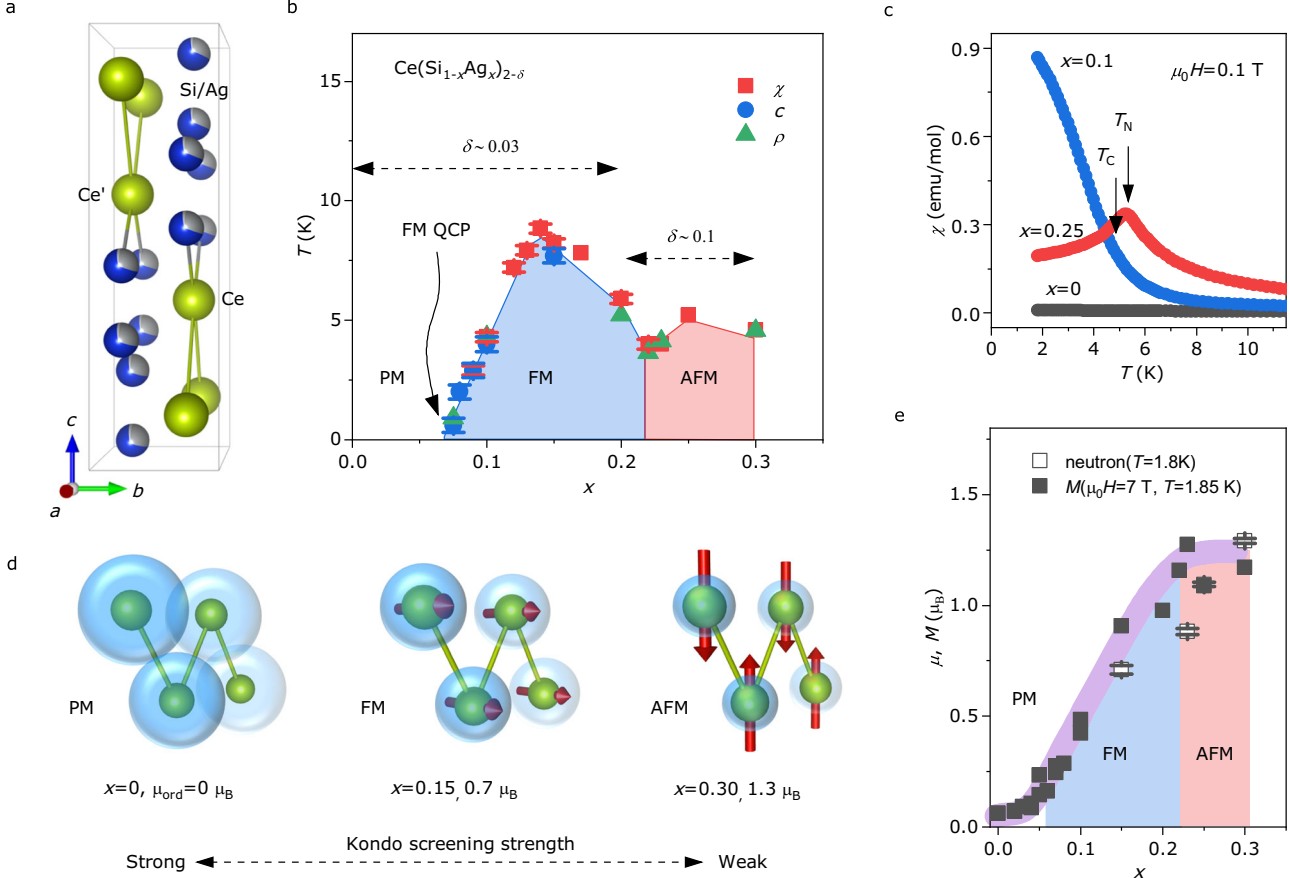

**Fig. 1 | Magnetic phase diagram and Kondo coupling of $Ce(Si_{1-x}Ag_x)_{2-\delta}$. a** Crystal structure of $Ce(Si_{1-x}Ag_x)_{2-\delta}$. Yellow, blue, and grey spheres represent Ce, Si, and Ag atoms, respectively. Ce and Ce' indicate two different Ce sublattices distinguished by different local atomic environments. **b** Magnetic phase transition temperatures, determined by measuring magnetic susceptibility $\chi$, specific heat capacity $C$, and electrical resistivity $\rho$, are plotted as a function of Ag-doping $x$. PM, FM, and AFM represent the paramagnetic, ferromagnetic, and antiferromagnetic states, respectively. **c** Magnetic susceptibility for PM, FM, and AFM states is plotted as a function of temperature. $T_C$ and $T_N$ indicate transition temperatures of FM and AFM. **d** Schematic drawings display magnetic structures of three different magnetic ground states and the strength of Kondo screening (translucid blue spheres). $\mu_{ord}$ indicates the size of the ordered magnetic moments determined by neutron scattering experiments in units of Bohr magneton $\mu_B$. **e** The ordered magnetic moment (open squares) and magnetisation value (closed squares) at $\mu_0H = 7$ T and $T = 1.85$ K are plotted as a function of $x$.

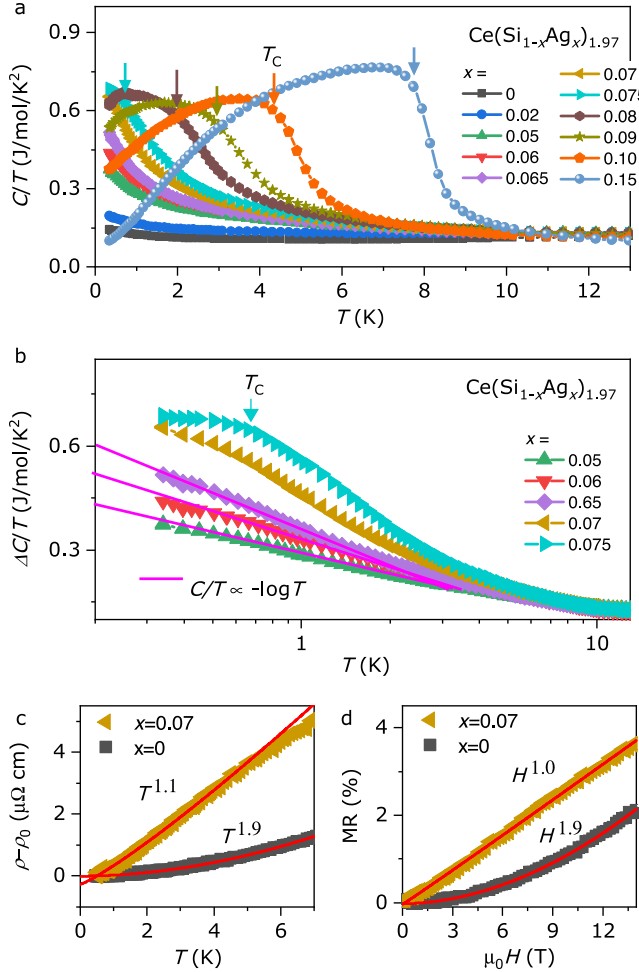

**Fig. 2 | Non-Fermi-liquid behaviour near the FM QCP of Ce(Si$_{1-x}$Ag$_x$)$_{1.97}$.**
**a** Specific heat capacity divided by temperature $C/T$ of Ce(Si$_{1-x}$Ag$_x$)$_{1.97}$ is plotted as a function of temperature. $T_C$ indicates the ferromagnetic transition temperature. **b** $\Delta C/T$, obtained from subtracting the $T^3$ term from $C$, is plotted as a function of temperature for $x$ near the critical concentration. The magenta line represents $C/T \propto -\log T$ at low temperatures. **c** Electrical resistivity $\rho$ after subtracting the residual resistivity $\rho_0$ for $x = 0$ and 0.07 is plotted as a function of temperature. The red lines are the least-squares fitting using $\rho - \rho_0 \propto T^n$. **d** Magnetoresistance MR for $x = 0$ and 0.07 were measured up to $\mu_0 H = 14$ T at $T = 1$ K. The red lines are the least-squares fitting using MR $\propto H^\beta$.

## Results

Figure 1b is a magnetic phase diagram of Ce(Si$_{1-x}$Ag$_x$)$_{2-\delta}$ showing various magnetic ground states, i.e., a paramagnetic (PM) state at $0 \le x < 0.07$, an FM state at $0.07 < x < 0.23$, and an AFM state at $0.23 \le x \le 0.30$. The transition temperatures were determined by kinks or peaks of the derivative of the magnetic susceptibility with respect to temperature $\partial\chi/\partial T$, electrical resistivity $\rho$, specific heat capacity $C$, and imaginary part of AC susceptibility $\chi_{AC}$ (for details, see Fig. S4 in Supplementary Information SI). Representative $\chi(T)$ curves of three magnetic ground states are plotted in Fig. 1c, measured using polycrystals of Ce(Si$_{1-x}$Ag$_x$)$_{2-\delta}$. FM and AFM transition temperatures are indicated by $T_C$ and $T_N$, respectively. In order to reveal the magnetic structures, neutron powder diffraction experiments were performed, and Rietveld refinements with symmetry analysis revealed that the Ce moments lie within a tetragonal plane in the FM state, as shown in Fig. 1d. On the other hand, ordered moments are perpendicular to the tetragonal plane in the AFM state (for details, see Figs. S5–S6 and Table S1 in SI). The variation in magnetic phases and the orientation of ordered moments depending on $x$ may be attributed to the

competition between FM and AFM exchange interactions and the degree of magnetic anisotropy. In principle, magnetic anisotropy can be behind the mechanism stabilising FM QCPs. If interactions along the $a$-axis are overwhelmingly larger than along the perpendicular directions, the FM QPT in Ce(Si$_{1-x}$As$_x$)$_{1.97}$ could be protected by the 1D character of the interactions, in analogy to the proposal for YbNi$_4$(P$_{1-x}$As$_x$)$_2$. However, the structural analysis shows that while the distances between Ce atoms in both $a$- and $c$-axis directions decrease with the reduction of Ag doping, their ratio remains almost constant, indicating that the structure retains its 2D character, making magnetic anisotropy an unlikely mechanism to protect the FM QCP in Ce(Si$_{1-x}$As$_x$)$_{1.97}$.

In the PM state, the absence of a magnetically ordered state is most likely due to the strong Kondo screening. Earlier inelastic neutron scattering experiments revealed that the strong Kondo effect in the PM state of CeSi$_{2-\delta}$ becomes weaker towards the FM state[17]. Indeed, the Kondo screening that reduces the magnitude of the localised moment becomes weaker with increasing $x$. As shown in Fig. 1e, the saturated field-dependent magnetisation at $\mu_0 H = 7$ T $M(\mu_0 H = 7T)$, by which all localised magnetic moments would be aligned along the external field direction, is increased with increasing $x$, indicating that the Kondo effect that screens the localised Ce moments[17] (blue spheres in Fig. 1b) is continuously suppressed by Ag-doping. Besides, the size of the ordered moments $\mu$ (empty squares) deduced from the neutron experiments is in line with the tendency of $M(\mu_0 H = 7$ T) (filled squares). The smaller $\mu$ compared with $M(\mu_0 H = 7T)$, could be attributed to the magnetic fluctuations or competing magnetic interactions.

Further increasing $x$, $M(\mu_0 H = 7T)$ is continuously increased until it reaches the saturated value of ~1.25 $\mu_B$ at $x \ge 0.22$, which is consistent with the expected average value of the localised moment calculated using the crystalline electric field (CEF) ground state doublet of CeSi$_{2-\delta}$, $\alpha|\pm5/2> + \beta|\mp3/2>$ [17] with $\alpha$ ~ 1. The CEF doublet of the FM state, CeSi$_{1.7}$, has $\alpha = \pm0.91$ and $\beta = \mp0.41$, and the CEF easy-axis is the $c$-axis, whereas the FM moment alignment is perpendicular to the $c$-axis. The hard-axis FMs in Kondo lattices have been understood by the interplay between soft particle-hole fluctuations and magnetic anisotropies[20]. Likewise, the in-plane FM moments of Ce(Si$_{1-x}$Ag$_x$)$_{2-\delta}$ could be associated with the CEF hard-axis due to strong fluctuations and small magnetic anisotropy. With increasing $x$, the degree of magnetic anisotropy may become stronger, giving rise to the easy-axis alignment in the AFM state. Further experimental work with single crystals is required to investigate the above scenarios.

Figure 2a shows that the peak in $C/T$ due to the FM transition decreases with decreasing $x$, as indicated by arrows, reaching $T = 0$ K near the critical concentration $x_c = 0.07$. Bulk property measurements do not show any feature of magnetic ordering down to 0.36 K at $x \le x_c$, indicating the FM QCP at $x_c$ ~ 0.07. As shown in Fig. 2b, large entropy accumulation is observed near $x_c$, with $C/T$ ~ 0.6 J/mol/K$^2$ at $T = 0.36$ K, comparable to other Ce-based heavy-fermion compounds[2,21]. $\Delta C/T$, obtained by subtracting the $T^3$ contribution from $C/T$ (for details, see Fig. S7 in SI), follows a $-\log T$ dependence, the hallmark of NFL behaviour, for $x = 0.06$ and 0.05, down to the lowest measured temperature. The NFL behaviour near $x_c$ is also supported by electrical transport data. Temperature-dependent electrical resistivity $\rho(T)$ of $x = 0$ and 0.07 are plotted in Fig. 2c. For $x = 0$, $\rho \propto T^2$ follows the standard FL theory, while for $x = 0.07$, $\rho \propto T$, another hallmark of NFL behaviour in the temperature range of $0.45$ K $< T < 5$ K. Furthermore, the magnetoresistance (MR) with the field applied to the out-of-plane direction, for $x = 0$ at $T = 1$ K exhibits an $H^2$ dependence, associated with normal-metal behaviour, whereas the MR for $x = 0.07$ shows an $H$-linear dependence up to $\mu_0 H = 14$ T. The linear MR behaviour has been reported for FeSe$_{1-x}$S$_x$ and La$_{2-x}$Ce$_x$CuO$_4$ due to quantum critical fluctuations[22,23]. The divergence of $C/T$, as well as $T$- and $H$-linear $\rho$ at $x$ ~ $x_c$, indicates that the FM QCP appears in the dense Kondo ferromagnet CeSi$_{1.97}$ by Ag-substitution.

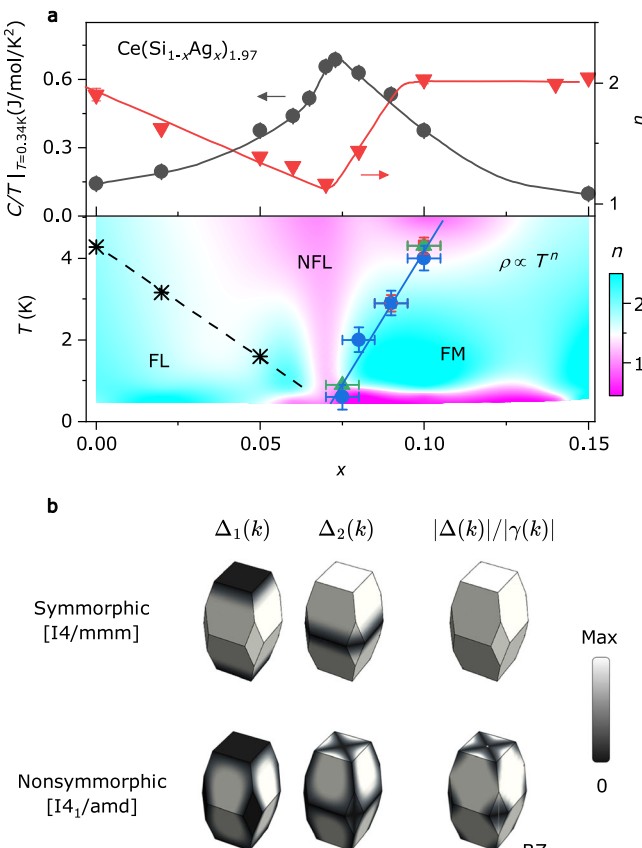

**Fig. 3 | Quantum criticality near the FM QCP protected by nonsymmorphic symmetry. a** In the upper panel, $C/T$ at $T = 0.34$ K and resistivity temperature exponent $n$, estimated by the least-squares fitting using $\rho(T) = \rho_0 + AT^n$, are plotted as a function of $x$ on the left and right ordinate, respectively. In the lower panel, $T_C$ and $T_{FL}$ are plotted as a function of $x$ overlaid on the colour plot of $n$ on the $T - x$ plane. **b** Normalised magnitude of the two components of inter-sublattice hopping (ISH), $\Delta(k) = (\Delta_1(k), \Delta_2(k))$, and ratio $|\Delta(k)|/|\gamma(k)|$, where $|\gamma(k)|$ corresponds to the magnitude of the spin-orbit coupling SOC plotted on the Brillouin zone (BZ) edges (the explicit momentum dependence of these functions is given in the SI). Note that this ratio goes to zero along some lines (black colour) due to nonsymmorphic symmetries (bottom), while it is not guaranteed to be zero in symmorphic systems (top). The dark grey colour around these lines indicates regions within which the SOC dominates over ISH.

Figure 3a summarises the quantum critical behaviour around the FM QCP in Ce(Ag$_{1-x}$Si$_x$)$_{1.97}$. The upper panel of Fig. 3a shows that $C/T$ at $T = 0.34$ K exhibits a peak at $x \sim x_c$. In addition, as plotted on the right ordinate in the upper panel of Fig. 3a, the exponent $n$ defining the temperature dependence of the resistivity, $\rho(T) = \rho_0 + AT^n$, is near unity, indicating a linear-in-temperature behaviour around $x_c$. The lower panel displays a colour plot of $n$, calculated from $\partial \ln(\rho - \rho_0)/\partial \ln T$ showing the expected funnel shape of NFL behaviour due to the magnetic quantum fluctuations, as reported in other FM and AFM QCPs[2,24]. The FL line in the phase diagram was obtained from the electrical resistivity measurements, below which $\rho(T)$ follows a $T^2$ dependence (for details, see Fig. S8 in SI).

## Discussion

Non-Fermi liquid behaviour can be observed in the absence of QCPs when metallic FMs are tuned by chemical doping[25,26] due to Griffiths phase effects[27]. Chemical doping induces inhomogeneity to the Kondo coupling strength, giving rise to Kondo-cluster-glass behaviour observed in CePd$_{1-x}$Rh$_x$[26] and CeNi$_{1-x}$Cu$_x$[25]. The scaling law of the quantum Griffiths phase is given by $\chi'_{AC} \propto C/T \propto T^{\varepsilon-1}$, $M \propto H^\varepsilon$, with

$0 < \varepsilon < 1$[28]. For instance, CePd$_{1-x}$Rh$_x$ exhibits the NFL behaviour without a diverging Grüneisen ratio, and experimental results are well explained by the above scaling laws[26]. However, Ce(Si$_{1-x}$Ag$_x$)$_{1.97}$ near $x_c$ exhibits a different scaling behaviour, with $\chi'_{AC} \propto T^{-1.7}$, $\Delta C/T \propto -\log T$, and $M \propto H^{0.5}$ (for details, see Fig. S9). The absence of Griffiths phase is further confirmed by the absence of hysteresis between zero-field-cooled and field-cooled $\chi(T)$ in the temperature range where the $\rho(T)$ exhibits a $T$-linear dependence ($T < 5$ K).

Recent studies of Si-substitution in the FM quantum critical matter CeRh$_6$Ge$_4$ suggested that impurities might have an important role in the phenomenology of the FM QCP[29]. In contrast to the NFL behaviour near the pressure-induced FM QCP, the electrical resistivity near the critical Si concentration does not exhibit the $T$-linear behaviour, although the Curie temperature seems to be suppressed continuously. At $x > x_c$, CeRh$_6$(Ge$_{1-x}$Si$_x$)$_4$ exhibits single-ion Kondo physics due to the inhomogeneous Kondo screening, which was not observed in Ce(Si$_{1-x}$Ag$_x$)$_{1.97}$. The $T$-linear $\rho(T)$ and the absence of inhomogeneous Kondo screening near $x \sim x_c$ suggest that the effects of the disorder are negligible in Ce(Si$_{1-x}$Ag$_x$)$_{1.97}$. Furthermore, in order to exclude the disorder effects, we plotted $T_C$ as a function of $(x - x_c)^{z_m v}$ in Fig. S10. Here, $z_m$ and $v$ represents the dynamical and correlation length exponents, respectively. The best scaling was obtained by $z_m v = 1$ with $x_c = 0.067$, similar to the one observed near the FM QPTs of YbNi$_4$(P$_{1-x}$As$_x$)$_2$[10] and CeRh$_6$Ge$_4$[2]. This linear scaling is different from the $z_m v \sim 2$ and 8/5 in asymptotic and pre-asymptotic regimes predicted by the disorder-induced QPTs[30]. Moreover, a sufficient amount of disorder is necessary to change the nature of the electronic soft modes from ballistic to long-range diffusive, thereby ensuring the FM QPT remains stable without reaching the tricritical point. Although it is not direct evidence, it is worth noting that the critical concentration of $x_c \sim 0.067$ in Ce(Si$_{1-x}$Ag$_x$)$_{1.97}$ is considered an insufficient amount of disorder compared with the disorder-induced QPTs, e.g., UCo$_{1-x}$Fe$_x$Ge ($x_c \sim 0.24$)[8], M$_{1-x}$Cr$_x$Si ($x_c \sim 0.2$)[9], and CePd$_x$Rh$_{1-x}$ ($x_c \sim 0.13$)[26].

Based on the experimental evidence supporting the FM QCP in Ce(Si$_{1-x}$Ag$_x$)$_{1.97}$, we now address the origin of its stability. Recent work by Kirkpatrick and Belitz[15] showed how inversion symmetry breaking in (globally) noncentrosymmetric materials can provide a loophole to avoid the general mechanism that transforms an FM QCP into a first-order phase transition. In noncentrosymmetric materials, antisymmetric spin-orbit coupling (SOC) splits the electronic bands in such a way that the two-particle excitations that couple to the magnetic order parameter acquire a mass and, therefore, do not contribute with non-analytic terms to the free energy. This proposal lays out a natural explanation for the observation of signatures of FM quantum criticality in the noncentrosymmetric heavy fermions CeRh$_6$Ge$_4$[2,11] and UIr[31]. Nevertheless, previous work by the same authors suggests that this loophole is valid only for materials that have global inversion symmetry breaking[32–34]. If the material is only locally noncentrosymmetric, an effective chiral symmetry guarantees the double degeneracy of the electronic structure and, therefore, the presence of soft modes that can couple to the magnetisation, leading to a first-order phase transition in the same fashion as in Fermi liquids without SOC. One notable exception to this conclusion is the case of systems for which extra lattice symmetries guarantee that the coupling between the distinct chirality sectors is zero[32], which makes the material behave effectively as two independent globally noncentrosymmetric systems. In the presence of terms that couple the two chiral sectors, the nonanalyticity is restored, but its strength is reduced by a factor of $|\Delta(k)|^2/|\gamma(k)|^2$ ($\Delta(k)$ corresponds to the coupling between the two chirality sectors and $\gamma(k)$ to the antisymmetric SOC)[33,34]. The calculations of Belitz and Kirkpatrick were done within the lowest order loop expansion[34]. Most recently, Miserev and collaborators[35] showed that second-order perturbation theory in the electron-electron interaction restores the nonanaliticities in the spin susceptibility and, therefore, the first-order nature of the FM transition in noncentrosymmetric systems. Even

**Table 1 | Summary of known heavy-fermion FMs displaying either a first or second-order phase transition**

| Material | Order | Space group | Magnetic anisotropy ($\chi_{easy}/\chi_{hard}$) | Inversion Glob./Loc. | Symmorphic | $D_{sub}$ |
|---|---|---|---|---|---|---|
| YbCu$_2$Si$_2$[45] | 1st | $I4/mmm$ | 3.3 | Y/Y | Y | 3.924 |
| YbIr$_2$Si$_2$[46] | 1st | $I4/mmm$ | 1.9 | Y/Y | Y | 4.032 |
| UGe$_2$[47] | 1st | $Cmmm$ | 3.2 | Y/N | Y | 3.854 |
| UCoGe*[48] | 1st | $Pnma$ | 11 | Y/N | N | 3.480 |
| URhGe*[48] | 1st | $Pnma$ | 5 | Y/N | N | 3.520 |
| UCoAl*[49] | 1st | $P\bar{6}_2m$ | 8 | N | Y | 3.394 |
| URhAl*[50] | 1st | $P\bar{6}_2m$ | 36.4 | N | Y | 3.625 |
| U$_3$P$_4$*[51] | 1st | $I\bar{4}_3d$ | 1.9 | N | N | 3.841 |
| CePt[52] | 2nd | $Cmcm$ | - | Y/N | N | 3.868 |
| CePd$_{1-x}$Ni$_x$[53] | 2nd | $Cmcm$ | - | Y/N | N | 3.873 |
| CeSi$_{1.81}$[19] | 2nd | $I4_1/amd$ | 1.9 | Y/N | N | 4.055 |
| CeRh$_6$Ge$_4$[2,11] | 2nd | $P\bar{6}m2$ | 220 | N | Y | 3.855 |
| UIr[31] | 2nd | $P2_1$ | 26 | N | N | 3.637 |
| YbNi$_4$(P$_{1-x}$As$_x$)$_2$**[10] | 2nd | $P4_2/mnm$ | 5.5 | Y/Y | Y | 3.857 |
| U$_4$Ru$_7$Ge$_6$**[54] | 2nd | $Im\bar{3}m$ | 1.4 | Y/N | Y | 4.144 |

We highlight the reported type of QCP, in addition to the space group of each material, drawing special attention to the presence or absence of global (Glob.) and local (Loc.) inversion symmetry and nonsymmorphicity. In addition, the last column ($D_{sub}$) gives the distances between sublattices of lanthanide or actinide sites, which trend corroborates the discussion in the main text. Exceptions of the proposed picture are highlighted by (*) and (**). YbNi$_4$(P$_{1-x}$As$_x$)$_2$ and U$_4$Ru$_7$Ge$_6$, with the second-order phase transition ascribed to the one-dimensional nature of the crystalline structure and a particularly large distance between inequivalent U sites relieving the need for nonsymmorphicity, respectively. Among the materials that display a first-order phase transition, exceptions to this rule include U(Co, Rh)Ge and U(Co, Rh)Al, which can be understood in terms of the small distances between sublattices, which enhances the ISH. Another exception is U$_3$P$_4$, which is globally noncentrosymmetric and nonsymmorphic, but displays a first-order phase transition. Despite the simple chemical formula, this system has a unit cell with exceptionally many atoms, which might make this simple picture not directly applicable.

though this result seems robust, the phenomenology presented here suggests that the prefactors accompanying such higher-order non-analiticities are much smaller than for Fermi liquids in the absence of SOC, resulting in a weaker effect, allowing for the experimental observation of NFL behaviour around putative FM QCP in these systems. In other words, there is no known mechanism that would always absolutely protect FM QCPs, but there are mechanisms that can suppress the temperature below which the first-order transition sets in. Occasionally, the first-order transition does not set in, characterising a legitimate FM QCP. If the first-order transition sets in at unobservable low temperatures, experiments would still observe the standard phenomenology of an FM QCP down to the lowest measured temperatures.

Motivated by these observations, we constructed Table 1. First, we note that, while the mechanism based on magnetic anisotropy is likely behind the protection of the FM QCP in CeRh$_6$Ge$_4$, other materials hosting a second-order phase transition have magnetic anisotropy ratios $\chi_{easy}/\chi_{hard}$ smaller than 50[36], suggesting that there might be a different mechanism behind the stabilisation of these FM QCPs. Remarkably, most of the heavy-fermion FMs that display a second-order phase transition and signatures of quantum criticality are either globally noncentrosymmetric, or locally noncentrosymmetric and nonsymmorphic. We, therefore, infer that nonsymmorphicity is a key ingredient in stabilising these FM QCPs. From a theoretical perspective, this can be understood from a symmetry analysis of the inter-layer processes in symmorphic and nonsymmorphic systems, as illustrated in Fig. 3b (see SI for more details). For symmorphic systems, the two components of $\Delta(\boldsymbol{k})$, corresponding to symmetric and antisymmetric inter-layer hopping processes, have zeros in different regions in momentum space, such that $|\Delta(\boldsymbol{k})|$ is never guaranteed to be zero by symmetry. In contrast, for nonsymmorphic systems, the two components of $\Delta(\boldsymbol{k})$ have matching regions in which they are zero, guaranteeing that the ratio $|\Delta(\boldsymbol{k})|^2/|\gamma(\boldsymbol{k})|^2$ can be made arbitrarily small if the Fermi surfaces are located close to the BZ boundaries (see Fig. 3b). A rough trend of the ratio $|\Delta(\boldsymbol{k})|^2/|\gamma(\boldsymbol{k})|^2$ can be captured by the atomic distances between inequivalent sublattices sites ($D_{sub}$, see last column in Table 1). The larger the inter-sublattice distances, the smaller the hopping amplitudes between the sublattices, which corresponds to a

weaker coupling between sectors with opposite chirality and potential protection of the FM QCP. For example, CePt, CePd$_{1-x}$Ni$_x$, CeSi$_{1.81}$, and UIr all display FM QCPs. These materials do not have large enough magnetic anisotropy but do have large inter-sublattice distances compared to U(Co,Rh)Ge and U(Co,Rh)Al, which display first-order phase transitions at zero temperature. For the particular case of CeSi$_2$, Ce and Ce' sublattices are not centres of inversion, despite the fact that the lattice is globally centrosymmetric. The centre of inversion lies at the midpoints between Ce and Ce'. Furthermore, the two sublattices are related by a nonsymmorphic symmetry−an in-plane mirror reflection followed by a half lattice vector translation along the $b$-axis− as can be seen in Fig. 1a (a more detailed discussion is given in the SI). The combination of these symmetries with a particularly large distance between the inequivalent sublattices might be at the core of the protection of the QCP against its transmutation into a first-order transition.

Nonsymmorphic symmetries have been regarded as a key ingredient for topological electronic band structures and phases. Our conjecture, therefore, suggests further studies on the interplay of topology and strong correlations and the role of topology on quantum critical phenomena. Research on these lines has recently been pushed in the context of Weyl Kondo semimetals[3] and twisted van der Waals structures[37]. Our work highlights nonsymmorphic metallic FMs as one new family of materials to be considered in this broader context. Furthermore, we believe the empirical guidelines presented here can be used as a general principle in the search for FM materials hosting QCPs, which could deepen our understanding of the necessary ingredients to sustain FM criticality. Lastly, FM fluctuations are known to be key for the nucleation of superconductivity with spin-triplet and possibly topological character. Doped CeSi$_2$ is, therefore, a promising material platform for the exploration of novel physics.

## Methods
### Material synthesis and characterisation
Polycrystalline samples of Ce(Si$_{1-x}$Ag$_x$)$_{2-\delta}$ were synthesised by the arc-melting technique[16]. Ce (rod, 99.9%, ChemPUR), Si (lump, 99.9999%, Alfa Aesar), and Ag (granule, 99.99%, ChemPUR) were prepared in a stoichiometric molar ratio. The weighted elements in stoichiometric

composition were melted several times after flipping over to improve the sample homogeneity. To improve the crystal quality, the arc-melted buttons covered by Ta-foil were sealed in an evacuated silica tube with ~7 Torr of Ar gas for thermal annealing at 850 °C for 10 days. All experimental results were obtained using polycrystalline samples. Phase purity was investigated by powder x-ray diffraction measurements (PXRD) using a Bruker D8 Advance with Cu-cathode. None of the cases show any impurity phases. Crystal structure and stoichiometry were investigated by single-crystal diffraction using an STOE STADIVARI diffractometer with Mo K$_\alpha$ radiation (0.71073 Å). For details of doping-dependent lattice parameters and chemical compositions, including all refinement results, see Tables S2–S5 in SI. All single-crystal x-ray diffraction resultswere measured using tiny pieces of single crystals obtained from polycrystalline samples. Although Ag and Si atoms possess chemically different characteristics, such as valence electrons and atomic radius, all neutron and x-ray diffraction patterns were successfully refined only when the doped Ag replaced Si. A similar experimental result concerning Ag substitution performed independently was reported[16] and another example of Ag-substitution on the Si-site was reported in Ba$_8$Ag$_x$Si$_{46-x}$ system[38]. Note that the nominal $x$ was used for $0 \le x \le 0.20$ because the Si-site deficiency is negligible and lattice parameters are systematically controlled with nominal $x$, while actual $x$ was used for a nominal value of $0.25 \le x \le 0.35$.

### Neutron powder diffraction and refinement
The crystal and magnetic structures were studied by neutron powder diffraction using the high-resolution powder diffractometer for thermal neutrons (HRPT)[39] at the Swiss Spallation Neutron Source SINQ at Paul Scherrer Institut (PSI), Switzerland. About 2 g of each powder sample was loaded in a 6 mm vanadium container. Diffraction patterns were collected at temperatures of $T$ = 1.8 and 15 K using neutrons with wavelengths of $\lambda$ = 1.494 and 2.45 Å. All diffraction data were analysed using the FullProf software suite[40]. The symmetry analysis of the magnetic structures was done using the Bilbao crystallographic server[41] and the ISODISTORT tool based on ISOTROPY software[42,43].

### Bulk property measurements
Electrical resistivity measurements were performed using the standard four-probe (25 μm Pt wires) technique, applying a current of 1 mA on the polished surface of bar-shaped specimens. Electrical resistance and heat capacity were measured by the physical property measurement system (PPMS, Quantum Design) with a He-3 insert. Magnetisation measurements were performed on a superconducting quantum interference device (SQUID) installed in the magnetic property measurement system (MPMS, Quantum Design), in the temperature and magnetic field ranges from $T$ = 1.8 to 300 K and $\mu_0 H$ = 0 to 7 T, respectively.

## Data availability
The experimental data used in this study are available in the Zenodo database under accession code 8363352[44].

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

## Acknowledgements

S.S. and E.P. would like to thank Eric D. Bauer and Hanoh Lee for the fruitful discussion. Neutron experiments were performed at SINQ at Paul Scherrer Institut, Switzerland. Electrical resistance, magnetisation, and heat capacity measurements were performed in Laboratory for Multi-scale Materials Experiments at Paul Scherrer Institut. This project has been supported by the Swiss National Science Foundation SNSF Project No. 200021_188706. AR acknowledges support from the Swiss National Science Foundation through Ambizione Grant No. 186043.

## Author contributions

S.S., A.R., and E.P. initiated and led the project. S.S. synthesised the samples and performed the bulk property measurements. S.S. and V.P. performed the neutron experiments and analysed the results. I.P. performed and analysed the single-crystal x-ray diffraction experiments. S.S., A.R., and E.P. wrote the manuscript with input from all authors.

## Competing interests

The authors declare no competing interests.
