## [Peer Review File · Nature Communications]

Ferromagnetic quantum critical point protected by nonsymmorphic symmetry in a Kondo metalREVIEWER COMMENTS

Reviewer #1 (Remarks to the Author):

The manuscript by Shin et al. reports experimental observation of the FM QCP in Ag doped CeSi₂, i.e., the linear temperature dependence of resistivity and the $-\log(T)$ dependence of the electronic specific heat. They attributed the presence of FM QCP to the spin-orbit coupling in a locally noncentrosymmetric crystal structure with nonsymmorphic symmetry, which ensures the analyticity in the spin susceptibility (hence the second-order nature of FM transition). The results provide valuable insight to understand the FM quantum criticality, which is of topical interest recently. In particular, the paper raises the interesting possibility that the nonsymmorphic crystal symmetry, which is often overlooked in strongly correlated systems such as heavy fermions (but is emphasized in the field of topological physics), can be important for realizing the FM QCP. The results are supported by careful experiments and theoretical analysis. Therefore, the paper opens up new opportunities to search for quantum critical phenomena and novel topological phases in correlated electron systems. I think that the paper should be suitable for publication in Nature Communications. However, I think that the authors need to address the following issues before it can be recommended for publication in Nature Communications:

1. Subtraction of the T^3 term in the specific heat. In Fig. 2(b), the authors mentioned the subtraction of a T^3 term in the specific heat and obtained the logarithmic divergence of the specific heat with temperature. How can the authors obtain (reliably) the contribution of such a T^3 term? It seems to me that no detailed information is given in the manuscript or SI. It is an important point and is a key experimental signature of FM QCP – therefore it needs to be clarified.
2. Effect of disorder. The presence of disorder (from doping) can often complicate the understanding of the FM QCP. For example, in the case of Si-substituted CeRh₆Ge₄, the impurity scattering could change the quantum critical behaviors. In the current case, the authors proposed that the Ag doping is primarily on the Si site. On the other hand, Ag and Si are chemically very different (with very different valence electrons and atomic radius). How can the authors rule out the possibility that the Ag doping does not take place on the Ce site? Does the Ag doping merely change the volume, or does it also change the electron count?
3. The local or itinerant type QCP. The authors provide a very useful summary of FM heavy fermion compounds with either first-order or second-order phase transition. From the

compiled list, the compounds with nonsymmorphic crystal structure indeed exhibit second-order FM transition for most cases, except for a few U-based compounds, where the quantum criticality is thought to be the “itinerant” type. Here the definition of the “itinerant” or “local” type should be more clear: why U-based compounds are considered to be “itinerant”, while Ce-based compounds are local? Is it due to more itinerant nature of 5f electrons? While I agree that 5f electrons are more delocalized compared to 4f electrons, strong Kondo effect (hence itinerant 4f quasiparticles) can still exist in Ce-based compounds. In fact, in the current case, CeSi₂ in the FM phase still show reduced ordered moment, implying appreciable Kondo screening or hybridization.

Other minor issues:

1. Table 1, line 3: presence “or” absence?

Reviewer #2 (Remarks to the Author):

Shin et al present an experimental study by magnetization, ac susceptibility, specific heat and resistivity of Ce(SiAg)_{2-d} compounds. They draw a doping temperature phase diagram. They find that the ground state is paramagnetic ground at low Ag doping, ferromagnetic with moments along a-axis for intermediate Ag doping, and antiferromagnetic with moments along c-axis at large Ag doping. A ferromagnetic quantum critical point at the end of a second order phase transition separates the paramagnetic and ferromagnetic phases at low temperature. The QCP is accompanied by a non Fermi liquid when the temperature is raised. Although the paper is experimental, there is a strong input from theory, and the main claim is related to a theoretical interpretation of a particular experimental result: the finding of a ferromagnetic QCP. The authors claim that Ag substitution in Ce(SiAg)_{2-d} leads to a unique ferromagnetic QCP driven by a change of the Kondo coupling, and that the second order of the quantum phase transition results from local noncentrosymmetry and nonsymmorphicity.

On one side, a subtle model is proposed to describe which conditions may favor a ferromagnetic QCP, which could be of interest for a wide audience. However, I feel that the

information about the basic magnetic properties, which can be extracted from the experimental data presented in the manuscript, is too limited and does not allow to test alternative scenarios. The possibility of alternative scenarios is also not mentioned in the paper. In particular, I think that we cannot exclude that a change of the magnetic properties, rather than a modification of the Kondo effect, could drive the QCP, and I am not convinced that the pertinence of the proposed model is fully demonstrated. For this reason, I do not think that this article may be of interest for a wide audience and I do not recommend it for a publication in Nature Communications. If the authors consider the comments listed below and made appropriate changes to the manuscript, mainly adding a few information and some nuances to their conclusions, I would recommend it for publication in Scientific Reports.

1- No indication about the direction of the magnetic field applied for the susceptibility and magnetization data is given. Such measurements with a field along the main directions of the crystal can inform about the evolution of the magnetic anisotropy as function of temperature and doping, and perhaps help extracting the crystal field ground state. It is not clear in the manuscript whether the field direction of magnetic properties measurement is not indicated because of the use of polycrystals or for another reason. If the authors can measure single crystals, I would suggest them to add magnetization and susceptibility data measured for the different main field directions (easy task using a MPMS or PPMS). They could try extracting the evolution of the magnetic anisotropy and of the electronic ground state as function of Ag doping, and discuss more quantitatively its relation with spin orbit coupling, whose role was considered in the theoretical scenario proposed by the authors, and the stabilization of a FM or AF order. I note that spin orbit coupling is known to be strong in Ce f electron systems and is generally associated with an energy/temperature scale of around 3000 K. On the other hand, a study made only on polycrystals would constitute a serious limitation to characterize their magnetic properties, and I would then suggest the authors to mention this clearly in the manuscript.

2- I do not agree with the claim by the authors that a reduced ferromagnetic moment is necessarily induced by the Kondo effect. In the vicinity of a ferromagnetic QCP, ferromagnetic fluctuations are strong and can also lead to a reduction of the ordered FM moment without implying a reduction of the moment by the Kondo hybridization between f and conduction electrons. AF fluctuations could also be responsible to the reduction of the ordered FM moment. I understand that the study of magnetic fluctuations (by neutrons or NMR) goes beyond the scope of this work, but such information would be needed before

concluding that magnetic moments are reduced only by the Kondo hybridization. I would suggest the authors to add nuances in their manuscript. They could for instance mention that they speculate that the reduction of the ordered moment is induced by the Kondo effect, but that a reduction of the ordered moment due to magnetic (FM or AF) fluctuations present near the QCP cannot be excluded.

3- As well as the evolution of the magnetic anisotropy mentioned above, it may be useful to consider the evolutions with Ag doping of the magnetic exchange interactions, which surely strongly influence the magnetic phase diagram. The manuscript nicely shows that Ag doping permits to tune the competition of FM and AF interactions. The presence of an AF phase at large Ag doping indicates thus that AF interactions are present in the system and may already compete with FM interactions at low Ag doping. This may be related with the reduced FM ordered moment. This competition is also accompanied by a change of the magnetic anisotropy, which is demonstrated by the neutron measurements presented in the manuscript: they indicate different directions of the magnetic moments in the FM and AF phases (FM moments ordered along a and AF moments ordered along c). This confirms that understanding the evolution of the magnetic anisotropy with Ag doping may be useful to describe the magnetic phase diagram of $\text{Ce}(\text{SiAg})_{2-d}$, including the FM and AF phases, and also the FM QCP. I would suggest the authors to mention these elements in the discussion, and at least to consider if they could lead to corrections or deviations (or not) to the theoretical description they propose for the FM QCP of $\text{Ce}(\text{SiAg})_{2-d}$.

4- in Figure 1, the authors mention that the dashed line indicates the expected value of $1.25 \mu\text{B}$ for Ce^{3+} moments. Could they precise how they obtained this value? Is it related to a given crystal field scheme? If yes how the crystal field ground state was determined?

Reviewer #3 (Remarks to the Author):

The authors reported a continuous ferromagnetic quantum phase transition (FM QPT) in $\text{CeSi}_{2-\delta}$ with Ag doping. Their measurements also revealed non-Fermi liquid

behaviors in the resistivity and specific heat at the QCP. Previously, a continuous FM QPT has been reported in pressurized CeRh6Ge4. The issue was interesting because earlier theory by Kirkpatrick and Belitz had predicted a first order transition. The theory was then revised to argue that noncentrosymmetric crystal structure with strong spin-orbit coupling might explain the continuous FM QPT in CeRh6Ge4. However, the idea was soon proved wrong.

Since CeSi_{2-\delta} is also centrosymmetric, the authors of this paper tried to argue that the combined influence of a local noncentrosymmetric structure with nonsymmorphic symmetry may preserve the stability of the FM QPT. Their idea is interesting, but we do not think that they have provided sufficient supports either experimentally or theoretically before they could clarify satisfactorily the following issues:

1) Continuous FM QPT is not rare in disorder systems. The merit of previous work in pressurized CeRh6Ge4 is that it was observed in a stoichiometric compound. In this work, the observation was made in CeSi_{2-\delta} by Ag doping. There are obviously even two types of disorder from both vacancy and doping. The authors excluded the Griffiths phase and inhomogeneous Kondo screening, but these are not necessarily present at all. We do not see strong direct evidences to exclude the possibility that the continuous FM QPT (or weakly first order QPT) may be due to the disorder effect.

2) The authors tried to infer that global noncentrosymmetry or local noncentrosymmetry and nonsymmorphicity is the key for the continuous FM QPT. Since noncentrosymmetry alone has been excluded, they proposed to consider local noncentrosymmetry and nonsymmorphicity. However, their collected data in Table 1 are quite diverse and do not seem to support such a conclusion. Rather, Table 1 negates their suggestion since all other situations are well possible.

3) Assuming their proposal is possible, they support their argument from Fig. 3c and the supplemental materials by showing the ratio $\Delta(k)/\gamma(k)$ vanishing along specific high symmetry lines on the Brillouin zone boundary. According to previous theory, nonanalyticity caused by soft modes should be proportional to this ratio. Could the authors confirm that the entire Fermi surfaces must lie on these high symmetry lines to avoid nonanalyticity? This seems impossible. Then how can nonanalyticity (hence first order

QPT) be avoided as the FM order parameter should be coupled to the whole Fermi surface rather than only some hot spots or high symmetry lines?

4) The manuscript does not clearly establish the key relation between nonsymmorphicity and the vanishing of the above-mentioned ratio, neither in the main article nor in the supplementary materials. Would this ratio be finite anywhere in the Brillouin zone if a material lacked nonsymmorphic symmetry? If not, would these materials also be able to host FM QPT? Clarification on this point is essential for future exploration of more examples.

5) For materials like $\text{YbNi}_4(\text{P}_{1-x}\text{As}_x)_2$ and CeRh_6Ge_4 , quasi-one dimensional (magnetic) structure, or the anisotropy of magnetic interaction (RKKY) along different directions, has been considered to be a crucial factor for its FM QCP. In particular, $\text{YbNi}_4(\text{P}_{1-x}\text{As}_x)_2$ is neither noncentrosymmetric nor nonsymmorphic. Could the authors briefly discuss this possibility for their material? From the variation of the magnetic orders with doping illustrated in Fig. 1b, it seems that reducing Ag doping leaves the interaction FM along one direction but tunes significantly the magnetic interaction along the other direction from AFM to FM. It is highly possible that further reducing Ag doping suppresses the magnetic interaction in one direction, hence causing strong anisotropy and the continuous FM QPT, which might then be related simply to the low dimensionality in the magnetic couplings but has nothing to do with noncentrosymmetry and nonsymmorphicity.

Reviewer #4 (Remarks to the Author):

Manuscript Number: NCOMMS-23-55223-T

Title: Ferromagnetic quantum critical point protected by nonsymmorphic symmetry in a Kondo metal

Reply to Reviewers' comments:

Reviewer #1 (Remarks to the Author):

The manuscript by Shin et al. reports experimental observation of the FM QCP in Ag doped CeSi₂, i.e., the linear temperature dependence of resistivity and the $-\log(T)$ dependence of the electronic specific heat. They attributed the presence of FM QCP to the spin-orbit coupling in a locally noncentrosymmetric crystal structure with nonsymmorphic symmetry, which ensures the analyticity in the spin susceptibility (hence the second-order nature of FM transition). The results provide valuable insight to understand the FM quantum criticality, which is of topical interest recently. In particular, the paper raises the interesting possibility that the nonsymmorphic crystal symmetry, which is often overlooked in strongly correlated systems such as heavy fermions (but is emphasized in the field of topological physics), can be important for realizing the FM QCP. The results are supported by careful experiments and theoretical analysis. Therefore, the paper opens up new opportunities to search for quantum critical phenomena and novel topological phases in correlated electron systems. I think that the paper should be suitable for publication in Nature Communications. However, I think that the authors need to address the following issues before it can be recommended for publication in Nature Communications:

Reply) We would like to thank the reviewer for acknowledging the novelty and recommending the publication of this work after addressing the comments.

(1) Subtraction of the T^3 term in the specific heat. In Fig. 2(b), the authors mentioned the subtraction of a T^3 term in the specific heat and obtained the logarithmic divergence of the specific heat with temperature. How can the authors obtain (reliably) the contribution of such a T^3 term? It seems to me that no detailed information is given in the manuscript or SI. It is an important point and is a key experimental signature of FM QCP – therefore it needs to be clarified.

Reply) The authors would like to thank the reviewer for pointing out the importance of showing how to estimate the T^3 -term of C/T . The T^3 -term of C/T was deduced from the least-squares fitting on C/T vs. T^2 at high temperatures, where it exhibits a linear behaviour. The examples of the fittings and subtractions are added to the supplementary information (Fig. S5), and relevant comments have been noted in the main text (in line 12 of page 5).

Fig. S5. T^2 -dependent heat capacity C/T and subtracted heat capacity $\Delta C/T$ of $\text{Ce}(\text{Si}_{1-x}\text{Ag}_x)_{1.97}$. Three panels show that representative T^2 -dependent subtracted heat capacity $\Delta C/T$ (open symbols), extracted from subtracting the linear term (red solid line) of T^2 -dependent C/T (closed symbols). The linear term was obtained by least-squares fitting of T^2 -dependent C/T at a high-temperature regime.

(2) *Effect of disorder.* The presence of disorder (from doping) can often complicate the understanding of the FM QCP. For example, in the case of Si-substituted CeRh_6Ge_4 , the impurity scattering could change the quantum critical behaviors. In the current case, the authors proposed that the Ag doping is primarily on the Si site. On the other hand, Ag and Si are chemically very different (with very different valence electrons and atomic radius). How can the authors rule out the possibility that the Ag doping does not take place on the Ce site? Does the Ag doping merely change the volume, or does it also change the electron count?

Reply) We confirmed that Ag replaces Si using neutron and x-ray diffraction experiments. As displayed in Extended Data Fig. E2, Table E1, Table E2, and Tables S1-S3 in SI, all diffraction data were successfully refined when the doped Ag replaced Si. The numbers in Extended Data Table E2 show that the lattice parameter linearly increases with increasing Ag-doping, and the volume change between $x = 0$ and 0.35 is around 7.4 %. We don't have evidence for the Ag-doping effects on the number of carrier density, and it can be a future study. To clearly address the Si-replacement by Ag, the sentence 'All neutron and x-ray diffraction patterns were successfully refined only when the doped Ag replaced Si.' has been added to the 'Methods – Materials synthesis and characterization' section (in line 15 of page 19).

(3) The local or itinerant type QCP. The authors provide a very useful summary of FM heavy fermion compounds with either first-order or second-order phase transition. From the compiled list, the compounds with nonsymmorphic crystal structure indeed exhibit second-order FM transition for most cases, except for a few U-based compounds, where the quantum criticality is thought to be the "itinerant" type. Here the definition of the "itinerant" or "local" type should be more clear: why U-based compounds are considered to be "itinerant", while Ce-based compounds are local? Is it due to more itinerant nature of 5f electrons? While I agree that 5f electrons are more delocalized compared to 4f electrons, strong Kondo effect (hence itinerant 4f quasiparticles) can still exist in Ce-based compounds. In fact, in the current case, CeSi₂ in the FM phase still show reduced ordered moment, implying appreciable Kondo screening or hybridization.

Reply) We thank the referee for highlighting that our compilation of FM heavy fermions is "very useful". We also thank the referee for raising the interesting question concerning the effects of "itinerant" versus "local" magnetism on the nature of the QCP. In the compilation of this table, we followed the notion of itinerant and local moment magnetism discussed in the review by M. Brando et al., Rev. Mod. Phys., 88, 025006 (2016): "We refer to systems where the conduction electrons are the sole source of the magnetization as "itinerant" ferromagnets, and to ones where part or all of the magnetization is due to localized spins as "localized"-moment ferromagnets." In this review, it is also discussed that the theory developed to address the nature of the QCPs is the same for both cases once a magnetic order parameter is defined and this coexists with electronic soft modes. As our analysis follows the same reasoning, we believe that

the “localised” or “itinerant” nature of the magnetic order does not affect the nature of the QCP. We compiled the information in this column in a preliminary exploration of the trends in these materials, but we have not used this feature in our analysis. For that reason, to avoid overwhelming the reader with extra information, we have decided to omit this column in the new version of the manuscript.

(4) Table 1, line 3: presence “or” absence?

Reply) We would like to thank the referee for finding the typo. We’ve replaced the “of” with “or”.

Reviewer #2 (Remarks to the Author):

Shin et al present an experimental study by magnetization, ac susceptibility, specific heat and resistivity of Ce(SiAg)_{2-d} compounds. They draw a doping temperature phase diagram. They find that the ground state is paramagnetic ground at low Ag doping, ferromagnetic with moments along a-axis for intermediate Ag doping, and antiferromagnetic with moments along c-axis at large Ag doping. A ferromagnetic quantum critical point at the end of a second order phase transition separates the paramagnetic and ferromagnetic phases at low temperature. The QCP is accompanied by a non Fermi liquid when the temperature is raised. Although the paper is experimental, there is a strong input from theory, and the main claim is related to a theoretical interpretation of a particular experimental result: the finding of a ferromagnetic QCP. The authors claim that Ag substitution in Ce(SiAg)_{2-d} leads to a unique ferromagnetic QCP driven by a change of the Kondo coupling, and that the second order of the quantum phase transition results from local noncentrosymmetry and nonsymmorphicity.

On one side, a subtle model is proposed to describe which conditions may favor a ferromagnetic QCP, which could be of interest for a wide audience. However, I feel that the information about the basic magnetic properties, which can be extracted from the experimental data presented in the manuscript, is too limited and does not allow to test alternative scenarios. The possibility of alternative scenarios is also not mentioned in the paper. In particular, I think that we cannot exclude that a change of the magnetic properties, rather than a modification of the Kondo effect, could drive the QCP, and I am not convinced that the pertinence of the proposed model is fully demonstrated. For this reason, I do not think that this article may be of interest for a wide audience and I do not recommend it for a publication in Nature

Communications. If the authors consider the comments listed below and made appropriate changes to the manuscript, mainly adding a few information and some nuances to their conclusions, I would recommend it for publication in Scientific Reports.

Reply) We would like to thank the referee for pointing out another possible scenario that should be considered when addressing the experimental results and for highlighting that our proposed model and mechanism for the protection of a ferromagnetic QCP “could be of interest for a wide audience”.

We politely disagree with the referee on his/her assessment of the suitability of our manuscript to Nature Communications. We firmly believe that our work will attract the attention of communities interested in the topics surrounding strong correlations, quantum criticality, topology, and superconductivity. Research in these directions has been recently pushed in the context of Weyl-Kondo semimetals and twisted van der Waals systems. Our work brings nonsymmorphic metallic ferromagnets as a new family of materials to be considered in this broader context, in which all these themes intertwine.

We believe the manuscript, revised according to the referee’s concerns and suggestions highlighting subtleties and alternative scenarios, has become more transparent and relevant for a broad audience.

(1) No indication about the direction of the magnetic field applied for the susceptibility and magnetization data is given. Such measurements with a field along the main directions of the crystal can inform about the evolution of the magnetic anisotropy as function of temperature and doping, and perhaps help extracting the crystal field ground state. It is not clear in the manuscript whether the field direction of magnetic properties measurement is not indicated because of the use of polycrystals or for another reason. If the authors can measure single crystals, I would suggest them to add magnetization and susceptibility data measured for the different main field directions (easy task using a MPMS or PPMS). They could try extracting the evolution of the magnetic anisotropy and of the electronic ground state as function of Ag doping, and discuss more quantitatively its relation with spin orbit coupling, whose role was considered in the theoretical scenario proposed by the authors, and the stabilization of a FM or AF order. I note that spin orbit coupling is known to be strong in Ce f electron systems and is generally associated with an energy/temperature scale of around 3000 K. On the other hand, a study made only on polycrystals would constitute a serious limitation to characterize their

magnetic properties, and I would then suggest the authors to mention this clearly in the manuscript.

- 1) Reply) All experimental results in the manuscript were obtained using polycrystalline samples of $\text{Ce}(\text{Si}_{1-x}\text{Ag}_x)_{2-\delta}$. In order to specify the information of the measured sample as the referee recommended, we have added the phrase "...measured using polycrystals of $\text{Ce}(\text{Si}_{1-x}\text{Ag}_x)_{2-\delta}$..." in the main text (in line 27 of page 3) and the sentence "All experimental results were obtained using polycrystalline samples" in the Methods section (Material synthesis and characterization, in line 8 of page 19). Note that the single crystal x-ray diffraction experiments were performed using the tiny pieces (~50 μm) of single crystals obtained from polycrystalline samples. Hence, in line 14 of page 19, in the Methods part, "..., measured using tiny pieces of single crystals obtained from polycrystalline samples." has been added.

As the referee suggested, the authors agree that investigating anisotropic magnetic properties using single crystals of $\text{Ce}(\text{Si}_{1-x}\text{Ag}_x)_{2-\delta}$ will deepen the understanding of the microscopic mechanism behind the stabilisation of and competition between antiferromagnetic and ferromagnetic orders. Thus, the authors have added the sentence in the main text (in line 5 of page 5), "Further experimental work with single crystals is required to investigate the above scenarios.

$\text{Ce}(\text{Si}_{1-x}\text{Ag}_x)_{2-\delta}$ single crystals can be synthesised by the Czochralski method, which was used for $\text{CeSi}_{2-\delta}$ single crystals and remains the only known method. The Czochralski method requires a lot of substance for each doping case and special equipment, which is currently unavailable. Instead, the flux method for $\text{Ce}(\text{Si}_{1-x}\text{Ag}_x)_{2-\delta}$, the best way to synthesise the series of single crystals, has been tried, but the attempts failed to get the proper phase. We would like to leave the single crystal study for future work to investigate the magnetic anisotropy and exact crystal field ground states as a function of Ag-doping. Nevertheless, we would like to point out that our polycrystal study, precisely revealing how the T_C was suppressed near the critical point by investigating the 12 doping cases, is clear evidence to demonstrate the existence of the FM QCP.

(2) I do not agree with the claim by the authors that a reduced ferromagnetic moment is necessarily induced by the Kondo effect. In the vicinity of a ferromagnetic QCP, ferromagnetic fluctuations are strong and can also lead to a reduction of the ordered FM moment without implying a reduction of the moment by the Kondo hybridization between f and conduction

electrons. AF fluctuations could also be responsible to the reduction of the ordered FM moment. I understand that the study of magnetic fluctuations (by neutrons or NMR) goes beyond the scope of this work, but such information would be needed before concluding that magnetic moments are reduced only by the Kondo hybridization. I would suggest the authors to add nuances in their manuscript. They could for instance mention that they speculate that the reduction of the ordered moment is induced by the Kondo effect, but that a reduction of the ordered moment due to magnetic (FM or AF) fluctuations present near the QCP cannot be excluded.

Reply) First, we would like to apologise for the confusion caused by using two terms, “localized moments” and “ferromagnetic ordered moments”, interchangeably. In the revised manuscript, we have clearly distinguished these terms such that the size of the “localized moments”, estimated from the saturated magnetization value at $\mu_0 H = 7$ T, is tuned by the Kondo screening, whereas the “ferromagnetically ordered moments” can be tuned by a combination of Kondo screening, magnetic fluctuations, and FM/AFM competing interactions. The authors would like to thank the reviewer for pointing this out. We have revised the comment accordingly, from line 20 to line 27 of page 4, “As shown in Fig. 1e, the saturated field-dependent magnetisation at $\mu_0 H = 7$ T $M(\mu_0 H = 7T)$, by which all localised magnetic moments would be aligned along the external field direction, is increased with increasing x , indicating that the Kondo effect that screens the localised Ce moments ¹⁷ (blue spheres in Fig. 1b) is continuously suppressed by Ag-doping. Besides, the size of the ordered moments μ (empty squares) deduced from the neutron experiments is in line with the tendency of $M(\mu_0 H = 7$ T) (filled squares). The smaller μ compared with $M(\mu_0 H = 7T)$, *e.g.*, $\mu = 0.71(2) \mu_B$, $M(\mu_0 H = 7T) = 0.91 \mu_B$ at $x = 0.15$, could be attributed to the magnetic fluctuations or competing magnetic interactions.” has been added.

To change the nuance, the relevant sentence in the Abstract, “We report that the Ag-substitution controls the strength of the Kondo coupling, leading to a transition between paramagnetic and ferromagnetic Kondo phases. Remarkably, a ferromagnetic QCP accompanied by concurrent strange-metal behaviour emerges.” has been revised to “We report that the Ag-substitution to CeSi_{1.97} linearly suppresses the ferromagnetic order to a QCP, accompanied by concurrent strange-metal behaviour.”

(3) As well as the evolution of the magnetic anisotropy mentioned above, it may be useful to consider the evolutions with Ag doping of the magnetic exchange interactions, which surely strongly influence the magnetic phase diagram. The manuscript nicely shows that Ag doping permits to tune the competition of FM and AF interactions. The presence of an AF phase at large Ag doping indicates thus that AF interactions are present in the system and may already compete with FM interactions at low Ag doping. This may be related with the reduced FM ordered moment. This competition is also accompanied by a change of the magnetic anisotropy, which is demonstrated by the neutron measurements presented in the manuscript: they indicate different directions of the magnetic moments in the FM and AF phases (FM moments ordered along a and AF moments ordered along c). This confirms that understanding the evolution of the magnetic anisotropy with Ag doping may be useful to describe the magnetic phase diagram of $Ce(SiAg)_{2-d}$, including the FM and AF phases, and also the FM QCP. I would suggest the authors to mention these elements in the discussion, and at least to consider if they could lead to corrections or deviations (or not) to the theoretical description they propose for the FM QCP of $Ce(SiAg)_{2-d}$.

Reply) The authors would like to thank the reviewer for pointing out the importance of magnetic anisotropy and competition between ferromagnetic and antiferromagnetic interactions in $Ce(Si_{1-x}Ag_x)_{2-\delta}$. Additionally, we noticed that the in-plane ferromagnetic ordering of $Ce(Si_{1-x}Ag_x)_{2-\delta}$ is a hard axis when the CEF ground doublet $|\pm 5/2\rangle + |\mp 3/2\rangle$ of $CeSi_{2-\delta}$ is considered. Including the hard axis possibility, the following discussion has been added in line 4 of page 4 “The variation in magnetic phases and the orientation of ordered moments depending on x may be attributed to the competition between FM and AFM exchange interactions and the degree of magnetic anisotropy.” and from line 29 of page 4 to line 6 of page 5 “The CEF doublet of the FM state, $CeSi_{1.7}$, has $\alpha = \pm 0.91$ and $\beta = \mp 0.41$, and the CEF easy-axis is the c -axis, whereas the FM moment alignment is perpendicular to the c -axis. The hard-axis FMs in Kondo lattices have been understood by the interplay between soft particle-hole fluctuations and magnetic anisotropies²⁰. Likewise, the in-plane FM moments of $Ce(Si_{1-x}Ag_x)_{2-\delta}$ could be associated with the CEF hard axis due to the strong fluctuation and small magnetic anisotropy. With increasing x , the degree of magnetic anisotropy may become stronger, giving rise to the easy axis alignment in the AFM state.”

(4) in Figure 1, the authors mention that the dashed line indicates the expected value of 1.25 μ_B for Ce^{3+} moments. Could they precise how they obtained this value? Is it related to a given crystal field scheme? If yes how the crystal field ground state was determined?

Reply) As the reviewer pointed out, there was no sufficient explanation about the crystalline electric field (CEF) scheme of the $4f^1$ orbital. In addition, we noticed that the sentence "...field-dependent magnetization $M(\mu_0H=7T)$ of $x = 0.1$ is $0.42 \mu_B$, indicating that the ordered Ce moments in the FM state are reduced to approximately 1/3 of Ce^{3+} moment." was misleading readers by suggesting that the crystalline electric field ground doublet was known. We have removed this sentence and added, "Besides, the size of the ordered moments μ (empty squares) deduced from the neutron experiments is in line with the tendency of $M(\mu_0H=7 T)$ (filled squares)." in line 21 of page 4 to better convey the intended meaning. To provide how we estimated the expected localized moment "... which is consistent with an expected average value of the localised moment calculated using the crystalline electric field (CEF) ground doublet of $CeSi_{2-\delta}$, $\alpha|\pm 5/2\rangle + \beta|\mp 3/2\rangle$ ¹⁷ with $\alpha \sim 1$." has been inserted in line 27 of page 4.

Reviewer #3 (Remarks to the Author):

The authors reported a continuous ferromagnetic quantum phase transition (FM QPT) in $CeSi_{2-\delta}$ with Ag doping. Their measurements also revealed non-Fermi liquid behaviors in the resistivity and specific heat at the QCP. Previously, a continuous FM QPT has been reported in pressurized $CeRh_6Ge_4$. The issue was interesting because earlier theory by Kirkpatrick and Belitz had predicted a first order transition. The theory was then revised to argue that noncentrosymmetric crystal structure with strong spin-orbit coupling might explain the continuous FM QPT in $CeRh_6Ge_4$. However, the idea was soon proved wrong.

Since $CeSi_{2-\delta}$ is also centrosymmetric, the authors of this paper tried to argue that the combined influence of a local noncentrosymmetric structure with nonsymmorphic symmetry may preserve the stability of the FM QPT. Their idea is interesting, but we do not think that they have provided sufficient supports either experimentally or theoretically before they could clarify satisfactorily the following issues:

Reply) The authors would like to thank the reviewers for commenting that the key idea of the manuscript is interesting and for pointing out the constructive concerns below. We believe that the revised manuscript addresses their concerns and provides sufficient experimental and theoretical evidence supporting our key message.

(1) Continuous FM QPT is not rare in disorder systems. The merit of previous work in pressurized CeRh6Ge4 is that it was observed in a stoichiometric compound. In this work, the observation was made in CeSi_{2-\delta} by Ag doping. There are obviously even two types of disorder from both vacancy and doping. The authors excluded the Griffiths phase and inhomogeneous Kondo screening, but these are not necessarily present at all. We do not see strong direct evidences to exclude the possibility that the continuous FM QPT (or weakly first order QPT) may be due to the disorder effect.

Reply) As the reviewer pointed out, it is critical to exclude the disorder effects in the stabilization of FM QCPs. The mechanism of the disorder-induced FM QPTs has been well explained by BKV (Belitz, Kirkpatrick, and Vojta) theory. The direct evidence we can provide is the scaling of $T_C \sim (x - x_c)^{z_m \nu}$ when z_m and ν represent the dynamical and correlation length exponent, respectively. BKV theory suggested that the critical temperature scaling follows $z_m \nu \sim 2$ and $8/5$ in asymptotic and pre-asymptotic regimes, respectively. The best T_C scaling of Ce(Si_{1-x}Ag_x)_{1.97} was obtained with $z_m \nu = 1$ with $x_c = 0.067$, as shown below, and this T_C scaling was also observed near FM QPTs of YbNi₄(P_{1-x}As_x)₂ and CeRh₆Ge₄. Moreover, sufficient disorder effects are necessary to change the nature of the electronic soft modes from ballistic to long-range diffusive, thereby ensuring the FM QPT remains stable without reaching the tricritical point. Although it is not direct evidence, it is worth noting that the critical concentration of $x_c \sim 0.067$ in Ce(Si_{1-x}Ag_x)_{1.97} is considered an insufficient amount of disorder compared with the disorder-induced QPTs, e.g., UC_{0.1-x}Fe_xGe ($x_c \sim 0.24$)⁸, M_{1-x}Cr_xSi ($x_c \sim 0.2$)⁹, and CePd_xRh_{1-x} ($x_c \sim 0.13$)²⁶. The above discussion has been added at the end of the second paragraph of the Discussion part (from line 22 of page 6 to line 3 of page 7), and the below graph has been added in the manuscript as Extended Data Fig. E5.

Extended Data Fig. E5. Critical exponent of T_c as a function of x . The best T_c scaling as a function of $(x - x_c)^{z_{mv}}$ was obtained by $z_{mv} = 1$ with $x_c = 0.067$. The dashed line represents the T_c scaling of the disorder-induced QPT in which $z_{mv} = 2$ at $0 \leq (x - x_c) < 0.05$ and $z_{mv} = 8/5$ at $0.05 \leq (x - x_c) < 0.08$.

(2) The authors tried to infer that global noncentrosymmetry or local noncentrosymmetry and nonsymmorphicity is the key for the continuous FM QPT. Since noncentrosymmetry alone has been excluded, they proposed to consider local noncentrosymmetry and nonsymmorphicity. However, their collected data in Table 1 are quite diverse and do not seem to support such a conclusion. Rather, Table 1 negates their suggestion since all other situations are well possible.

Reply) We politely disagree with the assessment of the referees on the fact that the compiled information in Table I negates our proposal that either global noncentrosymmetry or a combination of local noncentrosymmetry and nonsymmorphicity is key for the protection of FM QCPs.

First, we would like to comment on a previous statement by the referees, when they wrote that the revised Kirkpatrick-Belitz theory for noncentrosymmetric systems in the

presence of strong spin-orbit coupling was “proved wrong”. To our understanding, the calculations by Kirkpatrick-Belitz were not wrong. Within their considerations, the lowest order loop calculation indicated that the QCP is not transformed into a first order phase transition in the presence of a spin-split Fermi surface. Later work by the same authors discussed how, in the case of only local noncentrosymmetry, the tendency towards a first-order phase transition is controlled by the ratio $|\Delta(\mathbf{k})|^2/|\gamma(\mathbf{k})|^2$, which can be effectively understood in terms of the ratio t'/α between the inter-sublattice hopping (ISH), t' , and the strength of the spin-orbit coupling (SOC), α . Recent work by Miserev and collaborators has shown that considering higher order processes, one still finds nonanalyticities in the free energy, but again, these are smaller effects compared to the one expected from a spin degenerate Fermi surface proposed in the original work by Kirkpatrick and Belitz. What we want to emphasise here is that there is no known mechanism that would always absolutely protect FM QCPs, but there are mechanisms that can suppress the temperature below which the first order transition sets in. Occasionally, the first order transition does not set in, characterizing a legitimate FM QCP. If the first order transition sets in at unobservable low temperatures, experiments would still observe the standard phenomenology of a FM QCP down to the lowest measured temperatures. In our proposal, the mechanism relies on the presence of global noncentrosymmetry or the combination of local noncentrosymmetry and nonsymmorphicity. The above discussion has been newly added from line 30 of page 7 to line 4 of page 8.

Given the observations above, we now comment on how the compiled information in Table I is consistent with our proposal. Starting with the cases in which signatures of a second-order phase transition were observed experimentally, we see that they are primarily either globally noncentrosymmetric (CeRh₆Ge₄ and UIr), or locally noncentrosymmetric and nonsymmorphic (Ce(Pd,Pt)_{1-x}Ni_x, CeSi_{1.81}, UNiSi₂). The only exceptions are YbNi₄(P_{1-x}As_x)₂ and U₄Ru₇Ge₆. The former could be understood in terms of the one-dimensional nature of its crystalline structure. The latter could be understood by the fact that it is locally noncentrosymmetric and has the largest distance between f-electron sites among all listed materials, releasing the need of nonsymmorphicity to guarantee a small ISH/SOC ratio. Considering the cases in which signatures of a first-order phase transition were observed experimentally, the first material, Yb(Cu,Ir)₂Si₂ is globally and locally centrosymmetric and symmorphic, and the second material, UGe₂,

is locally noncentrosymmetric and symmorphic. Both materials fit within our proposal that either globally noncentrosymmetric or the combination of locally noncentrosymmetric and nonsymmorphic are necessary to protect the FM QCP. The exceptions are U(CoRh)Ge and U(Co,Rh)Al, the first locally noncentrosymmetric and nonsymmorphic, and the latter globally noncentrosymmetric. Naively, we would expect these materials to have a protected FM QCP. Note, however, that these have the smallest distances between f-electron sites, suggesting a large t'/α ratio that might not be enough to suppress the tricritical point to temperatures below the lowest experimentally measured temperatures. The case of U₃P₄ is more complex, as the unit cell is exceptionally large, and a more careful analysis might be needed. With this explicit discussion, which is also provided in the caption of Table I, we hope that we have convinced the referees that the data compiled in this table indeed supports our theoretical perspective.

(3) Assuming their proposal is possible, they support their argument from Fig. 3c and the supplemental materials by showing the ratio $\Delta(k)/\gamma(k)$ vanishing along specific high symmetry lines on the Brillouin zone boundary. According to previous theory, nonanalyticity caused by soft modes should be proportional to this ratio. Could the authors confirm that the entire Fermi surfaces must lie on these high symmetry lines to avoid nonanalyticity? This seems impossible. Then how can nonanalyticity (hence first order QPT) be avoided as the FM order parameter should be coupled to the whole Fermi surface rather than only some hot spots or high symmetry lines?

Reply) We thank the referees for this question, which allowed us to provide a more precise explanation of what we mean by the protection of the FM QCP against its transmutation into a first-order phase transition. As already discussed in answer to point (2) above, we do not claim that the combination of local noncentrosymmetry and nonsymmorphicity completely avoids the first-order phase transition, in the same way that it was shown that this is not the case for globally noncentrosymmetric systems. Nevertheless, we are confident that a small t'/α ratio does suppress the nonanalyticities in the free energy and, therefore, takes the tricritical point to lower temperatures, potentially making it not observable experimentally. Given these observations, our claim is not that the critical point is absolutely protected but that any signature of a first-order

phase transition is suppressed to very low temperatures if the ratio ISH/SOC is small. If the Fermi surfaces lie near the indicated black regions in Fig. 3c, this condition would most likely be satisfied. Unfortunately, the prediction of the Fermi surface shape and position for heavy fermion materials is a difficult task that challenges the best computational approaches. We cannot guarantee with certainty that the Fermi surfaces are around these regions. Nevertheless, our semi-quantitative analysis based on the trends on the 4f-site distances (last column in Table I) does indicate that our proposal is consistent with observations.

(4) The manuscript does not clearly establish the key relation between nonsymmorphicity and the vanishing of the above-mentioned ratio, neither in the main article nor in the supplementary materials. Would this ratio be finite anywhere in the Brillouin zone if a material lacked nonsymmorphic symmetry? If not, would these materials also be able to host FM QPT? Clarification on this point is essential for future exploration of more examples.

Reply) We thank the referees for raising this question, which allowed us to provide a more detailed explanation of the role of nonsymmorphic symmetry. The answer to the question “Would this ratio be finite anywhere in the Brillouin zone if a material lacked nonsymmorphic symmetry?” is yes. To show this explicitly, we have worked out the case of a structure with space group $I4/mmm$ (#139), the closest symmorphic analog of space group $I4_1/amd$ (#141). Both space groups are associated with body-centered tetragonal structures and are globally centrosymmetric. Taking a specific example, we focus on the ThGeSe-type structure. In this structure, all atomic sites are not at inversion centers, also characterizing it as locally noncentrosymmetric. Therefore, the main distinction between the ThSi₂- and the ThGeSe-type structure is that the first is nonsymmorphic, while the latter is symmorphic. The details of the structure are summarized in the new section, “Importance of nonsymmorphic symmetry,” in the Supplementary Information. From this explicit analysis, it becomes clear that the ratio ISH/SOC is never zero at any point in the Brillouin zone, as the two ISH parameters $\Delta_1(\mathbf{k})$ and $\Delta_2(\mathbf{k})$ vanish at distinct points of the Brillouin zone, see Figure 3 in the Supplemental Material. This is in contrast to what we have found for the nonsymmorphic structure, as was shown explicitly in Fig. 1 in the Supplementary Information.

(5) For materials like $\text{YbNi}_4(\text{P}_{1-x}\text{As}_x)_2$ and CeRh_6Ge_4 , quasi-one dimensional (magnetic) structure, or the anisotropy of magnetic interaction (RKKY) along different directions, has been considered to be a crucial factor for its FM QCP. In particular, $\text{YbNi}_4(\text{P}_{1-x}\text{As}_x)_2$ is neither noncentrosymmetric nor nonsymmorphic. Could the authors briefly discuss this possibility for their material? From the variation of the magnetic orders with doping illustrated in Fig. 1b, it seems that reducing Ag doping leaves the interaction FM along one direction but tunes significantly the magnetic interaction along the other direction from AFM to FM. It is highly possible that further reducing Ag doping suppresses the magnetic interaction in one direction, hence causing strong anisotropy and the continuous FM QPT, which might then be related simply to the low dimensionality in the magnetic couplings but has nothing to do with noncentrosymmetry and nonsymmorphicity.

Reply) As the reviewer pointed out, previous research suggests that the stabilization mechanism of the FM QPT in CeRh_6Ge_4 is associated with quasi-one-dimensional (1D) FM interactions (Ref. 2 in the manuscript). Similarly, the quasi-1D FM chain is pronounced in $\text{YbNi}_4(\text{P}_{1-x}\text{As}_x)_2$. As shown in Table 1 of the manuscript, we commented that the FM QPT of $\text{YbNi}_4(\text{P}_{1-x}\text{As}_x)_2$ can be understood through the characteristics of quasi-1D interactions. We recognise the lack of discussion on this stabilisation principle in the main text and address this shortfall by adding a new discussion in the context of $\text{Ce}(\text{Si}_{1-x}\text{As}_x)_{1.97}$. As suggested by the reviewer, with the reduction of Ag doping, the magnetic interactions along the c -axis have changed from AFM to FM. There are two ways to interpret this change. One is, as proposed, to view the magnetic interactions along the c -axis as being significantly reduced compared to those along the a -axis, resembling quasi-1D FM chains. If the 1D interactions along the a -axis are overwhelmingly larger in the FM state, it is possible that the FM QPT in $\text{Ce}(\text{Si}_{1-x}\text{As}_x)_{1.97}$, similar to $\text{YbNi}_4(\text{P}_{1-x}\text{As}_x)_2$, could also be greatly influenced by quasi-1D FM interactions. The other interpretation is that the interactions along the c -axis switch to FM, which would maintain 2D magnetic interactions. Structural analysis shows that while the distances between Ce atoms in both a - and c -axis directions decrease with the reduction of Ag doping, their ratio remains almost constant, indicating that the structure retains 2D character. Therefore, further studies, such as inelastic neutron scattering experiments, are necessary to investigate the anisotropy of Ce interactions as a function of Ag-doping. The protection of the FM QCP by quasi-1D interactions could potentially explain the

examples discussed above, but we believe it could not systematically address all materials compiled in Table I. Our motivation to consider lattice symmetries (global/local inversion symmetry and nonsymmorphic symmetries) comes then as a new proposal which could more broadly account for the phenomenology observed in these materials. The proposed principle of stabilization of the FM QPT through the symmetry of spin-orbit coupling not only applies to $\text{Ce}(\text{Si}_{1-x}\text{As}_x)_{1.97}$ but also provides an understanding of the trends for other cases presented in Table 1 of the manuscript. Thus, alongside the principle induced by the quasi-1D FM structure, we believe our proposal contributes significantly to FM QPT research. The relevant discussion has been added from line 6 to line 13 of page 4.

Reviewer #4 (Remarks to the Author):

Reply) The authors would like to thank the reviewer for their comments, which have allowed us to present our scientific findings in a more efficient and precise manner in the revised manuscript.

Summary of the changes made

(The changes made in the main text and supplementary information are highlighted in red font.)

(The line numbers addressed below are of the revised manuscript.)

- 2) In line 9 of the Abstract part, “We report that the Ag-substitution controls the strength of the Kondo coupling, leading to a transition between paramagnetic and ferromagnetic Kondo phases. Remarkably, a ferromagnetic QCP accompanied by concurrent strange-metal behaviour emerges.” has been revised to “We report that the Ag-substitution to $\text{CeSi}_{1.97}$ linearly suppresses the ferromagnetic order to a QCP, accompanied by concurrent strange-metal behaviour.”
- 3) In line 13 of page 3, “Fig. 1b” has been revised to “Fig. 1d”.
- 4) In line 17 of page 3, “... “and thermal expansion”...has been inserted.

- 5) In line 23 of page 3, “Fig. 1c” has been revised to “Fig. 1b”.
- 6) In line 24 of page 3, “ $0 \leq x < 0.1$, a FM state at $0.1 \leq x < 0.22$, and an AFM state at $0.23 < x \leq 0.30$ ” has been revised to “ $0 \leq x < 0.07$, a FM state at $0.07 < x < 0.23$, and an AFM state at $0.23 \leq x \leq 0.30$ ”
- 7) In line 27 of page 3, “Representative $\chi(T)$ curves of three magnetic ground states are plotted in Fig. 1c, measured using polycrystals of $\text{Ce}(\text{Si}_{1-x}\text{Ag}_x)_{2-\delta}$.” has been added.
- 8) In line 1 of page 4, “...symmetry analysis...” has been inserted.
- 9) In line 2 of page 4, “Fig. 1b” has been revised to “Fig. 1d”.
- 10) From line 4 of page 4 to line 14 of page 5, the discussion of the magnetic interactions and crystalline field effects has been revised.
- 11) In line 12 of page 5, ... (for details, see SI) ... has been inserted.
- 12) From line 22 of page 6 to line 10 of page 7, the discussion of the newly added Extended Data Fig. E5 has been added.
- 13) From line 30 of page 7 to line 4 of page 8, the discussion on the phenomenology of a FM QCP has been added.
- 14) In line 11 of page 8, “...(see last column in Table I). has been inserted.”
- 15) In line 17 of page 10, the reference 17 “Yashima, H., Satoh, T., Mori, H., Watanabe, D. & Ohtsuka, T. Thermal and magnetic properties and crystal structures of CeGe_2 and CeSi_2 . *Solid State Communications* **41**, 1-4 (1982). [https://doi.org/10.1016/0038-1098\(82\)90237-X](https://doi.org/10.1016/0038-1098(82)90237-X)” has been replaced with “Khogi, M., Satoh, T., Ohoyama, K., Arai, M. & Osborn, R. *Crystal field excitations in CeSi_x* . *Physica B* **163**, 137-140 (1990). [https://doi.org/10.1016/0921-4526\(90\)90149-Q](https://doi.org/10.1016/0921-4526(90)90149-Q)”.
- 16) In line 25 of page 10, the reference 20 “Sato, N., Mori, H., Satoh, T., Miura, T. & Takei, H. Thermal, Electrical and Magnetic Properties of the Ferromagnetic Dense Kondo System CeSi_x . *Journal of the Physical Society of Japan* **57**, 1384-1394 (1988). <https://doi.org/10.1143/JPSJ.57.1384>” has been replaced with “Krüger F., Pedder C.J., Green A.G., *Fluctuation-Driven Magnetic Hard-Axis Ordering in Metallic Ferromagnets*. *Physical Review Letters* **113**, 147001 (2014). <https://doi.org/10.1103/PhysRevLett.113.147001>”
- 17) In line 24, new reference 30, “Kirkpatrick, T.R., Belitz, D. *Quantum critical behavior of disordered itinerant ferromagnets*. *Physical Review B* **53**, 14364 (1996). <https://doi.org/10.1103/PhysRevB.53.14364>” has been added.
- 18) In line 4 of page 13, “...presence of absence...” has been replaced by “...presence or absence...”

- 19) We removed the column indicating the “Type” of QCP from Table I.
- 20) In Table I, information on the space group and symmorphicity of $\text{YbNi}_4(\text{P}_{1-x}\text{As}_x)_2$ has been corrected.
- 21) In the caption of Fig. 1, the last sentence “The dashed line indicates the expected value for Ce^{3+} of $\text{Ce}(\text{Si}_{1-x}\text{Ag}_x)_{1.9}$.” has been removed.
- 22) In line 8 of page 19, in the Methods part, “All experimental results were obtained using polycrystalline samples.” has been added.
- 23) In line 14 of page 19, in the Methods part, “..., measured using tiny pieces of single crystals obtained from polycrystalline samples.”
- 24) In line 15 of page 19, in the Methods part, “All neutron and x-ray diffraction patterns were successfully refined only when the doped Ag replaced Si.” has been added.
- 25) On page 26, a new figure, Extended Data Fig. E5, and its caption have been added.
- 26) A new section, “Importance of nonsymmorphic symmetry”, has been added to the Supplementary Information.
- 27) A new figure showing least-squares fittings of representative C/T at a high-temperature regime and the subtracted data $\Delta C/T$ have been added to the supplementary information.

Reviewers' comments:

Reviewer #1 (Remarks to the Author):

I have read the rebuttal letter from the authors carefully and I appreciate the authors' efforts for clarifying the relevant points and adding additional analysis. The manuscript is much improved. As I mentioned in my first report, the proposed idea of a FM QCP protected by nonsymmorphic symmetry + SOC in a locally noncentrosymmetric structure can be of broad interest to the community, if it can be supported by solid experimental/theoretical evidences. However, the polycrystalline nature of the samples with possible complication from substitution (by the way I am not completely convinced by statement of "the Ag substitution on the Si site") indeed limits the impact of the paper. Alternative explanations that include, e.g., dimensionality change of the magnetism, should not be overlooked. Therefore, I regret to say that this paper might be more suitable for publication in a more specialized journal, instead of Nature Communications.

Reviewer #2 (Remarks to the Author):

Shin et al made changes in their manuscript following remarks and recommendations from the three Reviewers, including part of my recommendations. This leads to an improvement of the manuscript. However, in continuity with my previous recommendation I do not recommend the publication of this work in Nature Communications. The authors did not convince me that their work brings a significant advance in the field, with strong experimental evidences for their conclusions, as expected from the standards of Nature Communications. In their resubmitted manuscript the authors mentioned that they have measured powder samples instead of single crystals (this information was missing in the manuscript submitted in the previous round). This constitutes a serious limitation for the conclusions about the studied electronic properties and to my opinion for the expected impact of this work. In the established literature on strongly correlated electrons, it is rare that a study on powder samples becomes of significant importance, and I think that the results presented here are not significant enough (there is no spectacular physical phenomenon as for instance unconventional superconductivity but just another quantum critical phase diagram reported here,) to become one of these rare cases. For these

reasons, I would recommend the publication of this work in Scientific Reports, which seems more appropriate for such work of high technical quality.

Reviewer #3 (Remarks to the Author):

Following my previous comments, the authors have made further analysis to rule out the role of disorder and given more detailed discussions on the relation between nonsymmorphicity and the ratio ISH/SOC . They admit that their proposed mechanism cannot fully suppress the nonanalyticity but only push the first order transition to lower temperatures. I think this point should be mentioned in the abstract, introduction, or discussion sections of the main article. In fact, given the complication of the Fermi surfaces, I personally don't think the suppression can be realized at all in real materials. For my questions 2 and 5, the authors' answers actually support the fact that the mechanism behind FM QCPs can be quite diverse. I'd like to mention that the RKKY coupling could be small as it changes character between AFM and FM, which may suppress the first order transition despite of the 2D crystal structure (hopping) [see Sci. China-Phys. Mech. Astron. 65, 257211 (2022)]. Nevertheless, the idea of introducing nonsymmorphicity is interesting and worthwhile of future investigations. I think the authors have tried their best to address all the questions. I therefore recommend the publication of the paper.

Reviewer #4 (Remarks to the Author):

Reply to Referee #1:

We thank Referee #1 for stating that the new version of our manuscript is much improved and that our work can be of broad interest.

We regret that the Referee has not been convinced about the nature of the Ag substitution, although we have provided strong experimental evidence. Rietveld refinement of diffraction patterns is the direct way to reveal the atomic positions and even occupancies if the contrast is good enough. As we have written in the manuscript, all our diffraction data can only be successfully refined when Ag replaces Si. As the Referee states, Si and Ag are chemically different in terms of valence electrons and atomic radius. However, as we cited in the manuscript, in Ref. 16, an independent analysis for $\text{Ce}(\text{Si}_{1-x}\text{Ag}_x)_{2-\delta}$ has already been published, indicating the same type of substitution ([https://doi.org/10.1016/S0925-8388\(00\)01474-2](https://doi.org/10.1016/S0925-8388(00)01474-2)). In addition, another Ag-substitution on the Si site has been reported for $\text{Ba}_8\text{Ag}_x\text{Si}_{46-x}$ (<https://doi.org/10.1103/PhysRevB.72.014511>). Therefore, our Rietveld refinement of neutron/x-ray diffraction results and the preceding examples strongly indicate that the added Ag atoms replace the Si atom in the CeSi_2 system. In order to stress this unusual doping, we have added the following statements to the ‘Method – Material synthesis and characterisation’ section on page 20, “Although Ag and Si atoms possess chemically different characteristics, such as valence electrons and atomic radius,...”, and “Similar experimental result concerning Ag substitution performed independently was reported ¹⁶, and another example of Ag-substitution on the Si-site was reported in $\text{Ba}_8\text{Ag}_x\text{Si}_{46-x}$ system ⁴⁸.”

We politely disagree that the polycrystalline nature of our samples limits the impact of our work. Firstly, experimental results with polycrystalline samples do not invalidate the existence of a ferromagnetic quantum critical point in the doped CeSi_2 system. Given the pressure-induced ferromagnetic quantum critical point investigated using the $\text{CeSi}_{1.86}$ single crystal (Ref. 19), our Ag-doping approach ensures the existence of the ferromagnetic quantum critical point in the CeSi_2 system by tracking the transition temperature down to 0.36 K and highlighting the associated non-Fermi liquid behaviour near the quantum critical point. Secondly, our theoretical proposal provides novel insight into the mechanisms behind the protection of ferromagnetic quantum critical points. Recent theory by J. Wang & Y.-F. Yang (Ref. 36) suggested the crucial role of strong magnetic anisotropy in stabilising ferromagnetic quantum critical points. As shown in the revised Table I of the manuscript, we investigated the magnetic anisotropy ratio of all compounds listed in the table, most of which are single crystals. We found that the magnetic anisotropy ratio cannot completely explain the robustness of ferromagnetic quantum critical points. Including CeSi_2 , most of the listed

compounds exhibiting a ferromagnetic quantum critical point do not satisfy the required anisotropy ratio, whereas they satisfy the trend suggested by our proposed mechanism.

Concerning the potential dimensionality change of magnetic interactions as a function of doping, we can provide the relevant experimental evidence from structural analysis. CeSi₂ exhibits an easy-plane magnetic anisotropy behaviour due to the crystalline electric field effects that are determined by the crystalline anisotropy. As discussed in the first paragraph of page 4, our structural analysis shows that while the distances between Ce atoms in both *a*- and *c*-axis directions decrease with the reduction of Ag doping, their ratio remains almost constant, indicating that the dimensionality of the magnetic interactions retains the easy-plane anisotropy as a function of Ag-doping.

Reply to Referee #2:

We thank Referee #2 for stating that the new version of our manuscript is improved.

We politely disagree with the Referee's opinion that our manuscript lacks "strong experimental evidence for our conclusions". We agree that, for the specific material reported, the polycrystalline nature of the samples is a limitation that cannot be overcome at the moment, as a synthesis of single crystals has not been successful. However, we are confident that our experimental results and the novel theoretical model will deliver significant advances to this field. Firstly, experimental results with polycrystalline samples do not invalidate the existence of the ferromagnetic quantum critical point in the doped CeSi₂ system. Given the pressure-induced ferromagnetic quantum critical point investigated using the CeSi_{1.86} single crystal (Ref. 19), our Ag-doping approach ensures the existence of the ferromagnetic quantum critical point in the CeSi₂ system by tracking the transition temperature down to 0.36 K and highlighting the associated non-Fermi liquid behaviour near the quantum critical point. Secondly, our theoretical proposal provides novel insight into the mechanisms behind the protection of ferromagnetic quantum critical points. Recent theory by J. Wang & Y.-F. Yang (Ref. 36) suggested the crucial role of strong magnetic anisotropy in stabilising ferromagnetic quantum critical points. As shown in the revised Table I of the manuscript, we investigated the magnetic anisotropy ratio of all compounds listed in the table, most of which are single crystals. We found that the magnetic anisotropy ratio cannot completely explain the robustness of ferromagnetic quantum critical points. Including CeSi₂, most of the listed compounds exhibiting a ferromagnetic quantum critical point do not satisfy the required anisotropy ratio, whereas they satisfy the trend suggested by our proposed mechanism.

Our theoretical proposal and its corroboration by sixteen different materials do not seem to be taken into account by the Referee. We strongly believe that our new experimental results, in light of the bigger picture provided by all the materials listed in Table I, provide an appealing scenario to explain the stabilisation of ferromagnetic quantum critical points based on crystalline symmetries.

Referee #2 also states that "there is no spectacular phenomenon as, for instance, unconventional superconductivity." We would like to take on this comment to actually highlight the value and potential impact of our work. Unconventional superconductivity with triplet character is of great interest to the scientific community given its exotic nature and potential applications. Our theoretical proposal provides a new guideline for the search for such superconductors, which are usually stabilised in the vicinity of ferromagnetic quantum critical points.

We also would like to stress that we have clearly mentioned that our samples are polycrystalline in the "Methods – material synthesis and characterisation" section of the initial submitted manuscript.

Reply to Referee #3:

We thank Referee #3 for the positive comments on the revised version of our manuscript.

We would like to highlight that the statement concerning the fact that the proposed mechanism cannot completely suppress the non-analyticity but only suppress it is already stated in the main body of the manuscript in the "Discussion" section on pages 7/8.

We thank the Referee for highlighting the potential role of magnetic anisotropy in stabilising ferromagnetic quantum critical points and the corresponding reference (newly added in the revised manuscript as Ref. 36). As suggested in this reference, CeRh₆Ge₄ exhibits a strong magnetic anisotropy and the protection of its ferromagnetic quantum critical point can be understood by this mechanism. Given this recent theoretical finding, we have revised Table I of the manuscript to include the magnetic anisotropy ratio of all listed compounds and added the comment "First, we note that, while the mechanism based on magnetic anisotropy is likely behind the protection of the FM QCP in CeRh₆Ge₄, other materials hosting a second-order phase transition have magnetic anisotropy ratios $\chi_{\text{easy}}/\chi_{\text{hard}}$ smaller than 50³⁶, suggesting that there might be a different mechanism behind the stabilisation of these FM QCPs.", and "For example, Ce(Pd, Pt)_{1-x}Ni_x, CeSi_{1.81}, UNiSi₂, and UIr all display FM QCPs. These materials do not have large enough magnetic anisotropy but do have large inter-sublattice distances compared to U(Co,Rh)Ge and U(Co,Ge)Al, which display first-order phase transitions at zero temperature." in the second paragraph of page 8.

Summary of the changes made

(The changes made in the main text and supplementary information are highlighted in red font.)

- 1) In line 5 of page 3, we have inserted the following phrase: "... also hosts a FM QCP..."
- 2) From line 5 to line 9 of page 8, we have added the following sentence: "First, we note that, while the mechanism based on magnetic anisotropy is likely behind the protection of the FM QCP in CeRh₆Ge₄, other materials hosting a second-order phase transition have magnetic anisotropy ratios $\chi_{\text{easy}}/\chi_{\text{hard}}$ smaller than 50³⁶, suggesting that there might be a different mechanism behind the stabilisation of these FM QCPs."
- 3) In line 14 of page 8, (see Fig. 3c) has been inserted.
- 4) From line 17 to line 21 of page 8, we have inserted the following sentences: "and potential protection of the FM QCP. For example, Ce(Pd, Pt)_{1-x}Ni_x, CeSi_{1.81}, UNiSi₂, and UIr all display FM QCPs. These materials do not have large enough magnetic anisotropy but do have large inter-sublattice distances compared to U(Co,Rh)Ge and U(Co,Ge)Al, which display first-order phase transitions at zero temperature."
- 5) On page 12, the new reference 36 has been added: Wang, J. & Yang, Y.-F. A unified theory of ferromagnetic quantum phase transitions in heavy fermion metals. *Science China Physics, Mechanics & Astronomy* **65**, 257211, (2022). <https://doi.org/10.1007/s11433-022-1879-2>
- 6) On page 13, the new reference 48 has been added: Kamakura, N. *et al.* Role of Ag doping in Ba₈Si₄₆ compounds. *Physical Review B* **72**, 014511 (2005). <https://doi.org/10.1103/PhysRevB.72.014511>
- 7) On page 15, the new column 'Magnetic anisotropy ratio ($\chi_{\text{easy}}/\chi_{\text{hard}}$)' has been inserted.
- 8) In line 15 of page 20, we have inserted the following phrase: "Although Ag and Si atoms possess chemically different characteristics, such as valence electrons and atomic radius,..."
- 9) From line 18 to line 20 of page 20, we have added the following sentence: "A similar experimental result concerning Ag substitution performed independently was reported¹⁶, and another example of Ag-substitution on the Si-site was reported in Ba₈Ag_xSi_{46-x} system⁴⁸."

REVIEWERS' COMMENTS

Reviewer #1 (Remarks to the Author):

I thank the authors for clarifying the issues related to Ag substitution and adding discussions to table I, which definitely improve the manuscript. I believe that the authors have tried their best to address the relevant issues. At this stage, whether this paper is suitable for publication in Nature Communications is probably a matter of personal taste.

On one hand, the proposed mechanism of FM QCP protected by nonsymmorphic symmetry is very interesting and can be of broad interest to the community (as I have already pointed out in the first round of review). With a more complete compilation and analysis of the FM QCP cases (in table I), as well as open discussions on other relevant models on FM QCP, the impact of the current work is further enhanced. On the other hand, the polycrystalline nature of the Ag-doped CeSi₂ samples in the current case will undermine the impact in some degree, as defects and other complications inherent in polycrystalline samples often hinder in-depth understanding of the underlying mechanism (unless single crystals become available for future studies).

With all these in mind, I would personally be fine to recommend its publication in Nature Communications. Firstly, the proposed mechanism of nonsymmorphic symmetry for the FM QCP adds new dimension to the research of FM quantum criticality, which is currently of topical interest. In particular, the nonsymmorphic symmetry is often connected to the nontrivial topology in the electronic states, which can open up new research directions. Secondly, the proposed mechanism can be applicable to other materials systems, as listed in table I, and not necessarily limited in Ag-doped CeSi₂. This can stimulate future experimental efforts. Finally, as theory becomes a very important part of the paper, I would suggest that the authors move some important parts of the theory in the supplementary information to the main text (e.g., page 8, second paragraph), to highlight the novelty in the theoretical part. This can help appeal to a broader audience.

Reviewer #2 (Remarks to the Author):

I maintain my recommendation not to publish the manuscript of Shin et al in Nature Communications, for the same reasons than those detailed in my previous reports.

In response to Referee #1's recommendations, we have revised Table 1 to ensure full compliance and thorough analysis of the ferromagnetic quantum phase transition cases. Additionally, we have revised Fig. 3 to more clearly illustrate the theoretical component of our work, incorporating relevant discussion as outlined in lines 1 to 8 on page 7.